# LARGE LANGUAGE MODELS ARE INTERPRETABLE LEARNERS

**Ruochen Wang**[*]
UCLA

**Si Si**
Google

**Felix Yu**
Google

**Dorothea Wiesmann**
Google

**Cho-Jui Hsieh**
Google, UCLA

**Inderjit Dhillon**
Google

## ABSTRACT

The trade-off between expressiveness and interpretability remains a core challenge when building human-centric models for classification and decision-making. While symbolic rules offer interpretability, they often lack expressiveness, whereas neural networks excel in performance but are known for being black boxes. This paper shows a combination of Large Language Models (LLMs) and symbolic programs can bridge this gap. In the proposed LLM-based Symbolic Programs (LSPs), the pretrained LLM with natural language prompts provides a massive set of interpretable modules that can transform raw input into natural language concepts. Symbolic programs then integrate these modules into interpretable decision rules. To train LSPs, we develop a divide-and-conquer approach to incrementally build the program from scratch, where the learning process of each step is guided by LLMs. To evaluate the effectiveness of LSPs in extracting interpretable and accurate knowledge from data, we introduce IL-Bench, a collection of diverse tasks, including both synthetic and real-world scenarios across different modalities. Empirical results demonstrate LSP's superior performance compared to traditional neurosymbolic programs and vanilla automatic prompt tuning methods. Moreover, as the knowledge learned by LSP is a combination of natural language descriptions and symbolic rules, it is easily transferable to humans (interpretable), and other LLMs, and generalizes well to out-of-distribution samples. Our code and benchmark will be released for future research.

## 1 INTRODUCTION

Learning interpretable predictive models from annotated data remains a key challenge in human-centric AI. Given input-output pairs $\{(x_i, y_i)\}$, the objective is to learn a function $f : x \rightarrow y$ that not only fits the data accurately but is also interpretable. In this context, a strong form of "interpretable" means that human with no prior domain knowledge can understand and apply the decision rules demonstrated by $f$, facilitating *the transfer of knowledge from AI to humans*. This is crucial not only for enhancing the transparency of AI systems but also for enabling humans to learn from these models, empowering various human-in-the-loop applications such as scientific discovery, material synthesis, and automatic data annotation (Chaudhuri et al., 2021).

**Definition 1.1** *A predictive model is considered interpretable if its decision rules can be understood and applied by a human judger without prior domain knowledge.*

Consider an exemplar task of classifying species in Palworld (Pair, 2024) - a newly released Pokemon-style game - based on a few image-label pairs, as illustrated in Figure 1. The ultimate goal is that even humans unfamiliar with Palworld can replicate AI's decisions by following the same predictive rules after examining the model trained on the data. This task effectively represents the challenge of extracting interpretable knowledge, such as species characteristics, from data. The algorithm we propose in this paper learns a model following the decision rule illustrated in Figure 1, which is designed to be easily understood and reproduced by humans. In essence, this problem can be viewed as discovering interpretable knowledge (e.g., the properties of a species in Palworld) from the data.

---

[*]Work completed during internship at Google.

Despite extensive research, the problem of developing a fully interpretable predictive model has not been fully addressed. Traditional methods often face a trade-off between expressiveness and interpretability: Deep neural networks, for instance, are powerful yet operate as "black boxes". Although post-hoc explanation methods attempt to make these models more transparent by identifying influential features (Zintgraf et al., 2017; Petsiuk et al., 2018; Dabkowski & Gal, 2017; Shrikumar et al., 2017; Sundararajan et al., 2017; Ancona et al., 2017), they do not clarify the underlying decision-making processes and have no control over the learning process. Directly learning interpretable models like (locally) linear (Ribeiro et al., 2016), tree-based (Lundberg, 2017) often falls short in expressiveness, especially with complex inputs like images.

To address this challenge, **Neurosymbolic Programs (NSPs)** (Chaudhuri et al., 2021; Shah et al., 2020; Cui & Zhu, 2021; Nauta et al., 2021b) offer a promising solution by modeling the decision rule as a program incorporating both symbolic operations and neural network modules. Despite this, the inherent trade-off between expressiveness and interpretability persists. While the integration of neural modules enhances expressiveness, it also compromises the program's overall interpretability. Additionally, designing effective symbolic operators requires significant expertise and is critical for the performance of the resulting program, necessitating careful customization for each specific dataset (Chaudhuri et al., 2021; Shah et al., 2020; Cui & Zhu, 2021).

Is it possible to harness the power of neural networks within Neurosymbolic Programs without compromising interpretability? This paper presents an affirmative answer. Our key insight is that (Multimodal) LLMs encompass a variety of powerful, conditional probabilistic sub-models. These models share a unified parametric architecture with the unconditional parent LLM (Super Model), yet distinctive defined by their respective prompts. Therefore, crafting prompts (by either Human or meta-LLMs) for LLM is equivalent to searching over the hypothesis space spanned by these submodels. This yields an infinite set of neural network-based operations that are inherently interpretable and can serve as fundamental "learnable" building blocks within Neurosymbolic Programs.

Building on this insight, we introduce a novel framework termed **LLM-Symbolic Programs (LSPs)**, defined and learned through LLMs. Our approach leverages a minimal Domain-Specific Language (DSL) set with only two operators: prompted-LLM and conditional branching, yielding a classic decision-making process structured as trees. We then propose a learning algorithm to incrementally learn the tree using LLMs with prompt optimization. To thoroughly evaluate the efficacy of LSPs, we construct the **Interpretable-Learning-Benchmark** of diverse predictive tasks, containing both synthetic and real-world data across vision and text modalities. Our empirical findings show that LSPs surpass the accuracy of both traditional XAI methods and LLMs prompted with automatically learned instructions, all while maintaining human interpretability. These results highlight the potential of LSPs to significantly enhance the performance and utility of Multimodal LLMs in various applications.

## 2 BACKGROUND AND RELATED WORK

**Taxonomy** Interpretable learning (IL) is a central aspect of Explainable AI (XAI). The taxonomy closely follows that of discriminative tasks: for a given dataset $(x, y)$, the objective is to construct a model that not only predicts accurately but also provides insight into its predictions. Here, the knowledge required for making accurate predictions is not inherent to the model; rather, it must be distilled from the data into compact, interpretable rules. In this work, we use a strong form of "interpretability" defined as follows:

**Traditional IL methods** The pursuit of interpretable model predictions divides into two primary methodologies: post-hoc and intrinsic. Post-hoc methods explain the behavior of pre-trained models by identifying salient features, yet they fall short of fully recovering the neural decision-making process. In contrast, intrinsic methods, such as Neuro-Symbolic Programming (NSP) (Chaudhuri et al., 2021; Shah et al., 2020; Cui & Zhu, 2021; Nauta et al., 2021b), integrate interpretability directly into the model architecture. However, NSP faces a fundamental trade-off between expressiveness (requiring more neural network modules) and interpretability (favoring symbolic modules). Additionally, training NSP models is often computationally expensive due to the need for co-optimizing both program architecture and neural network parameters (Shah et al., 2020; Cui & Zhu, 2021).

**Interpretable Learning in the era of (M)LLMs** The vast corpus of knowledge encoded during the web-scale pretraining of (M)LLMs has empowered (M)LLMs with remarkable zero-shot capabilities

across diverse tasks, including math, coding, creative writing, etc. **However, IL tasks pose a unique challenge for these models, as they are inherently not zero-shot solvable** (Table 1). Specifically, LLMs must utilize knowledge acquired from labeled examples rather than relying solely on input data and its prior knowledge (including external knowledge retrieved via RAG).

*(1). Can existing prompting methods apply to IL tasks?* Most LLM prompting methods, such as Tree-of-Thoughts (Yao et al., 2024) or augmenting LLMs with various tools (calculator, symbolic solver, etc) (Dong et al., 2023; Fang et al., 2024; Yang et al., 2023b), do not involve any learning and are thus incompatible with IL tasks. Generic Prompt Optimization (PO) methods, which aim to automatically configure instructions for LLMs, could be applied to any task, including IL in principle (Zhou et al., 2022; Pryzant et al., 2023; Yang et al., 2023a; Singh et al., 2023; Wang et al., 2023). However, PO methods are predominantly designed for instruction induction task - inferring optimal task descriptions - rather than extracting concrete predictive rules from data (Zhou et al., 2022; Zhang et al., 2023). Consequently, most PO approaches focus on rewriting prompts to enhance performance (Pryzant et al., 2023; Hsieh et al., 2023), which is insufficient for deriving interpretable knowledge from data. Additionally, while recent developments have introduced capabilities for correcting prompts using error examples (Pryzant et al., 2023; Wang et al., 2023), they remain inadequate for extracting complex decision rules, such as conditional branching required for classification. These rules, often applicable to only a subset of samples, are challenging to recover when considering the entire training set. Our experiments show that directly applying existing methods fails to effectively address these complex decision rules. These limitations motivate the proposed LSP framework, which integrates prompt optimization with symbolic programs to overcome these challenges.

*(2). Can existing benchmarks measure (M)LLM's IL ability?* Despite the extensive study of IL in the pre-LLM era, there lacks of benchmarks suitable for evaluating such methods on modern (M)LLMs. Traditional XAI Datasets are often image-centric and inadequate for evaluating the text capabilities of LLMs. Furthermore, the inclusion of popular vision datasets like CUB within MLLM training corpuses leads to data contamination, making it difficult to determine if performance improvements are due to enhanced rule learning or mere retrieval of prior knowledge. LLM Benchmarks, such as Big-Bench (Suzgun et al., 2022), SuperNatural Instructions (Wang et al., 2022), and Math datasets (Cobbe et al., 2021; Trieu & Luong, 2024; Wei et al., 2024), measures various language ability of the model, ranging from prompt optimization, reasoning tasks, to summarization. However, all these tasks are all zero-shot solvable, allowing LLMs to make predictions without additional rule learning. Therefore, these benchmarks are unsuitable for evaluating IL tasks.

A Comprehensive literature review on the previous XAI methods, Neuro-Symbolic Programming, and Prompt Optimization methods can be found in Appendix A.1.

| | Interpretable Learning | Common LLM tasks |
|---|---|---|
| Zero-shot solvable? | ✗ – Solving the task requires extracting rules from labeled training data. | ✓ – LLMs can in principle solves these tasks without seen any labeled examples. |
| Representative tasks | Palword classification; Symbolic classification tasks | Big-Bench-Hard, Abstract Reasoning, Math, Coding, Agent, Summarization, RAG. |
| Example data | **Input** which creature in the Palworld-dex is this? **Output:** creature_1 | **Input:** Do you return to the starting point? Take 8 steps. Turn around. Take 8 steps. **Output:** Yes |

Table 1: Comparison between the taxonomy of Interpretable Learning and common LLM tasks.

# 3 IL-BENCH: 1ST INTERPRETABLE-LEARNING BENCHMARK FOR (M)LLMS

To address the lack of suitable benchmarks for evaluating the interpretable learning capabilities of (M)LLMs, we introduce the **I**nterpretable-**L**earning Benchmark (IL-Bench). This new benchmark comprises a series of challenging tasks that are not solvable through zero-shot methods by even the most advanced (M)LLMs, such as GPT-4 and Gemini-1.5. IL-Bench includes 16 new symbolic and real-context tasks unseen to the current model lineup. These tasks range across vision and language modalities, providing a comprehensive and extensible evaluation framework. Below, we provide a high-level summary of the key data curation methods; Concrete examples, data curation, statistics, and how to extend this benchmark can be found in the Appendix A.2 (Table 8).

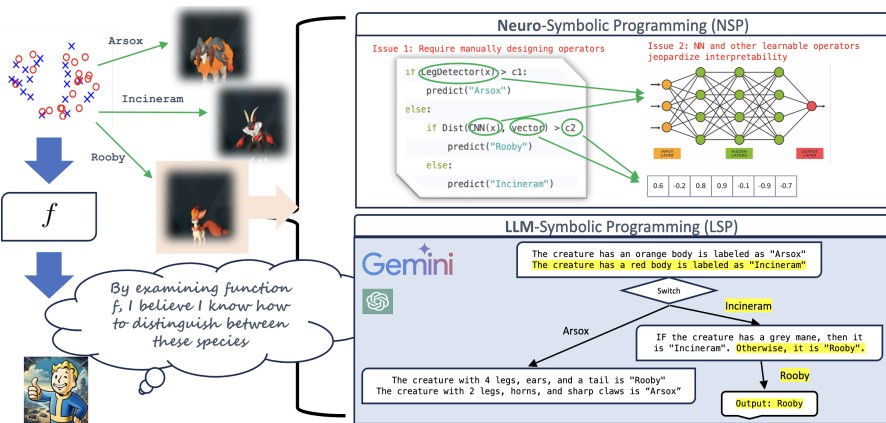

Figure 1: **Illustration of LLM-Symbolic vs. Neuro-Symbolic Program on interpretable learning task.** The goal is to develop a model that allows humans with no prior knowledge to replicate AI's decisions by following the same rules as the model. While NSP (Top right) offers a certain level of interpretability, it heavily relies on manually designing operators, and the inclusion of neural operators often reduces interpretability. In contrast, LSP (Bottom right) generates fully interpretable programs with the help of versatile LLM modules.

**Symbolic tasks**  Drawing inspiration from language-independent IQ tests, we generate set of synthetic datasets to evaluate the interpretable learning capabilities of the models. These datasets utilize symbols to denote input variables and their values; The input values are randomly assigned, and mapped to their labels based on a predefined set of rules (See Figure 8 for a concrete example). We also vary the number of variables, values, and labels to generate datasets of increasing complexity. These symbolic tasks enjoy several key benefits: (1). *Known oracle rules*, enabling precise evaluation of learning ability. (2). *Context independence*, forcing the models to depend solely on learned rules, without relying on external context. (3). *Scalability*, allowing for the automated creation of an unlimited number of tasks with arbitrary difficulty levels.

**Textual classification tasks: converting vision dataset to text inputs**  To evaluate model proficiency in intricate real-world scenarios, we utilize Fine-Grained Visual Classification (FGVC) datasets (Maji et al., 2013; Wah et al., 2011; Kramberger & Potočnik, 2020; Nilsback & Zisserman, 2008; Van Horn et al., 2015), such as CUB commonly used in XAI research. These datasets comprise of objects within narrowly-defined, visually-similar categories that are particularly challenging for the model to distinguish. To adapt these visual datasets for textual evaluation, we convert them into text-based datasets using a captioning model. In order for the task to be well-defined, the generated caption must cover all visual features required for classification, which are usually very subtle for FGVC datasets (e.g. the particular shape of a bird's beak). To ensure the captions capture all essential visual features, we also provide contrastive examples to the captioner (details in Appendix). The class names (e.g. Sea_Albatross) are also anonymized by symbols (e.g., class_1) to prevent the model from using label names to "shortcut" the prediction process. Empirical results indicate that the performance of existing text-based LLMs approximates that of random guessing in zero-shot setting.

**Visual classification Tasks: distinguishing novel visual concepts**  Due to the extensive coverage of (M)LLM training data, evaluating models in a multi-modal setting presents a unique challenge. Despite our best efforts, all existing image classification datasets we tested were already seen by at least one (M)LLM, which can predict labels in a zero-shot manner. To address this, we curate seven new datasets using screenshots from "Palworld," a recently released regional game featuring various creature species similar to Pokémon (examples in Table 8). As this game was released after the knowledge cut-off dates of the tested (M)LLMs, the models lack prior information about these creatures, requiring them to rely solely on the knowledge extracted from the dataset for predictions.

## 4  INTERPRETABLE LEARNING WITH LLM-SYMBOLIC PROGRAMMING

This section explains our proposed framework: LLM-Symbolic Programs. Section 4.1 reviews Neurosymbolic Learning method. Section 4.2 discusses utilizing LLM to implement interpretable programs, including a connection between prompted-LLM and interpretable unit (Section 4.2.1), the Domain Specific Language (Section 4.2.2) and learning algorithm (Section 4.2.3).

### 4.1 PRELIMINARIES ON CLASSICAL NEUROSYMBOLIC LEARNING

NeuroSymbolic Programming (NSP) (Chaudhuri et al., 2021; Shah et al., 2020; Cui & Zhu, 2021; Frosst & Hinton, 2017) represents an innovative method for combining classical symbolic learning with contemporary neural networks, with the goal of building expressive and interpretable models. NSP often consists of two main components: (1) a **Domain Specific Language (DSL)** that specifies available operations of the program (akin to a "search space") and (2) a **learning algorithm** for finding the best program. The resulting programs are structured, neuro-symbolic terms that follow the syntax specified by the DSL.

**Domain-Specific Language (DSL)** DSL in NSPs comprises manually defined operators, including interpretable symbolic (e.g. `if-then-else`) and expressive neural components (e.g. `cnn(x, θ)`). These operators can be chained to construct various tree-structured programs, a.k.a. computation graphs. equation 1 presents an example DSL used to construct the program for predicting the creature species in Figure 1. Here, $x$ and $c$ represents inputs and constants, and $\alpha$ denotes a sub-program:

$$\alpha = x \mid c \mid \text{Add}(\alpha_1, \alpha_2) \mid \text{Mul}(\alpha_1, \alpha_2) \mid \text{If } \alpha_1 \text{ Then } \alpha_2 \text{ Else } \alpha_3 \mid \text{cnn}(x, \theta) \mid \text{Dist}(\alpha_1, \alpha_2). \quad (1)$$

**Co-optimization of program structure and learnable parameters** In NSPs, the construction of a program involves solving a combinatorial optimization problem for both the program structure and the parameters of its learnable operators (e.g. neural components). As the number of DSL operators increases, the complexity of this task grows exponentially. To make the search process more tractable, existing research employs various approximation techniques to efficiently identify viable candidates, including greedy tree search (Shah et al., 2020), continuous relaxation (Cui & Zhu, 2021), distillation (Frosst & Hinton, 2017) and meta-learning (Chaudhuri et al., 2021).

**Limitations** While the integration of symbolic and neural components in NSPs represents a promising innovation, the incorporating of neural modules inevitably introduces black-box components and makes the program non-interpretable. Researchers have attempted to address this issue through two primary approaches: restricting the DSL to only interpretable operators (Shah et al., 2020; Cui & Zhu, 2021), or employing prototype learning to derive relatively interpretable neural modules (Nauta et al., 2021b; Ming et al., 2019; Nauta et al., 2021a). However, the DSL approach is not automatic, heavily relies on domain expertise, and potentially overlooking crucial information not identified by experts; Conversely, prototype learning aims to represent the concept of each neural module by a set of representative samples, which is not guaranteed to success.

### 4.2 LLM-SYMBOLIC PROGRAMS

This section explores how LLMs can effectively be utilized to implement NSPs' modules that are expressive, interpretable, and straightforward to learn with LLMs.

#### 4.2.1 PROMPTED-LLM AS AN INTERPRETABLE UNIT

The trade-off between interpretability and expressiveness presents a fundamental limitation in machine learning. Machines perceive images and text as raw binary signals, and transforming these into interpretable concepts; this inevitably requires complex and non-interpretable components, such as neural networks. Even human perception remains non-interpretable, as we lack a complete understanding of how the brain processes signals. However, the following analysis suggests that pretrained LLM offer a potential avenue to bridge this gap: it shows that powerful LLM can be used to define a wide range of interpretable functions via prompting.

**Connection between interpretable learning and prompting** LLMs pretrained on the next-token prediction task model the following joint distribution of a sequence of tokens $\{w_t\}_{t=1}^T$

$$P(w_1, w_2, \ldots, w_T) = \prod_{t=1}^T P(w_t \mid w_{t-1}, w_{t-2}, \ldots, 1) = f_\theta(w_t \mid w_1, w_2, \ldots, w_{t-1}),$$

where the conditional probabilities are parameterized by an auto-regressive model $f(\cdot; \theta)$ (e.g. Transformer) and each word $w_t$ is predicted given all the preceding tokens. The pretraining objective minimizes the following negative log-likelihood:

$$\min_\theta \mathcal{L}(\theta) = -\sum_{t=1}^T \log f_\theta(w_t \mid w_{t-1}, \ldots, w_1). \quad (2)$$

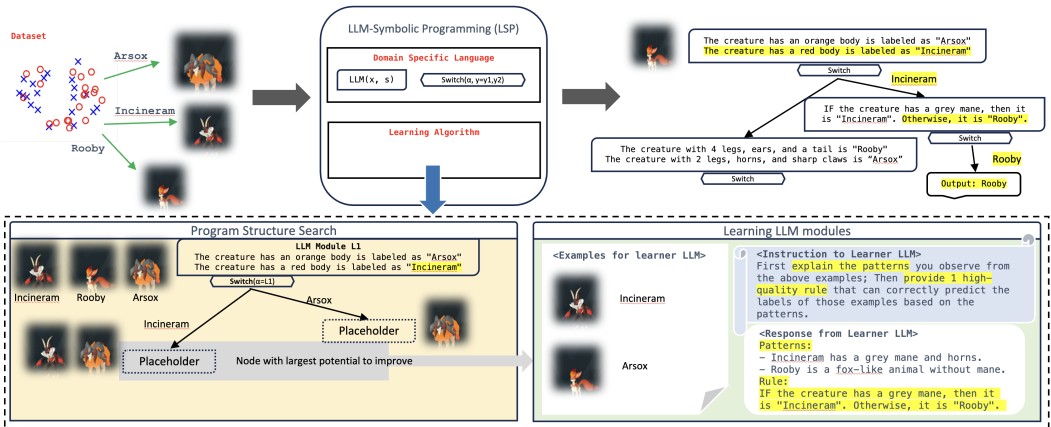

Figure 2: **Learning Algorithm for LSPs.** The learning algorithm for LSPs contains two parts: (1) program structure search (Left): This process is akin to constructing a traditional decision tree. Starting from the root, the algorithm traverses down the tree, iteratively splitting the training dataset based on the current node's predictions and expanding the leaf node with the highest prediction errors. (2) LLM module optimization (Right): Here, a learner LLM is instructed to summarize rules based on the observed data at its node.

A key observation from Eq. equation 2 is that the training process optimizes a "SuperNet" of conditional probabilistic models (CPM), each defined by an instruction $s$: $f_{s,\theta}(y|x) = f_\theta(y \mid x, s)$, where $x$ is the input and $s$ is the instruction for a particular task. Therefore, with a fixed LLM, the set of natural language prompts, denoted as $\mathcal{S}$, provides a massive set of interpretable neural network modules for the task. For a given dataset $\{(x_i, y_i)\}_{i=1}^n$, finding the best prompt to minimize the empirical loss, $\min_{s \in \mathcal{S}} \sum_{i=1}^n \mathcal{L}((f_{s,\theta}(y_i \mid x_i)))$, can be viewed as a form of learning, and the resulting model is inherently interpretable, as the prompt $s$ is expressed in natural language.

This connection reveals that prompt within the natural language space offers a form of interpretable learning that simultaneously achieves both expressiveness and interpretability. The key to bridging this gap lies in leveraging LLMs to handle the non-interpretable processing of raw signals into high-level concepts, much like how neurons in the human brain transform signals into information. This allows learning to occur within an interpretable space.

### 4.2.2 DOMAIN-SPECIFIC LANGUAGE OF LSPS

Traditional NSPs require manually designing a comprehensive DSL. However, with LLM's ability to represent a wide range of functions via different prompts, we can significantly streamline the grammar required to build expressive and interpretable models. Specifically, for predictive models, we can build powerful LSPs from a minimalist DSL with only three components: the input, conditional branching, and LLM module:

$$\alpha ::= x \mid \mathtt{switch}(\{\alpha == y_i : \alpha_i\}_{i=1}^k) \mid \mathtt{LLM}(x, s). \tag{3}$$

Here, **input** $x$ represents the input data (text, image, etc); the **conditional branching** $\mathtt{switch}(\{y_i : \alpha_i\}_{i=1}^k)$ forms the backbone of the program structure. Each switch can be viewed as a node in a decision tree tree with $k$ branches. It will branch to $\alpha_i$ if the sub-program $\alpha$ predicts $y_i$. **The LLM Module** $\mathtt{LLM}(x, s)$ serves as the inference engines. It means to prompting LLM to make a prediction on input $x$ under the instruction $s$.

Figure 1 (Bottom Right) shows an example program generated from above DSL. During inference time, given a test query, we traverse the tree-structured program in a top-down manner, assigning data to specific child node based on the parent node's predictions, until the leaf node is reached and the final response is returned.

### 4.2.3 LEARNING ALGORITHM

After defining the search space for program construction, we proceed to describe the algorithm used to identify the optimal program. Similar to Neuro-Symbolic Programming (NSP), our approach involves optimizing two key components:

- *LLM module optimization*: Generating the rules from data for each LLM module.
- *Program structure search*: Determining how to expand the program tree.

Figure 2 illustrates the entire search process. The following sections will describe these two components respectively.

**LLM modules optimization via summarizing predictive rules**  In Large Symbolic Programs (LSPs), each LLM module is responsible for making decisions on its designated data subset. While traditional NSPs optimize neural modules through empirical risk minimization, LSPs can derive predictive rules directly from observed data, a method we termed **RuleSum**. To achieve this, we leverage the LLM's powerful summarization capabilities (Adams et al., 2023; Goyal et al., 2022; Zhang et al., 2024; Pu & Demberg, 2023), and instruct a learner LLM to observe patterns from the data samples and summarize them into concrete rules. The process is visualized in Figure 2 (right).

**Program Structure Search**  LSP produces a tree-structured program where each path represents a complete decision-making process. To discover the optimal program, we employ a top-down tree traversal approach to expand the tree from scratch. Starting from the root node of an empty program with the entire training dataset:

- *Step 1:* Add an `LLM(x, s)` module to the root node.
- *Step 2:* Optimize `LLM(x, s)` using the **RuleSum** algorithm.
- *Step 3:* Create child nodes for the root by adding a `switch` operator to the program.
- *Step 4:* Assign training data to child nodes based on `LLM(x, s)`'s predictions.
- *Step 5:* Move to the highest-scoring child node, and repeat Steps 1–4 until `max_iter` is reached.

In essence, this search algorithm uses a divide-and-conquer strategy: it progressively partitions the training dataset into sub-branches based on the parent node's predictions, enabling the child LLM modules to further refine the prediction. This approach simplifies the learning process for each LLM module and makes the overall system more error-tolerant: the RuleSum algorithm only needs to derive rules for a subset of the data, and any inaccuracies can be corrected by subsequent child nodes.

**Node scoring function for node selection**  During program structure search, we prioritize the expansion of the node with the highest potential for program improvement. Since nodes with a higher frequency of errors have greater room for enhancement, we use error count as the scoring function. This metric, which considers both the error rate and the size of the data subset handled by each node, offers a straightforward yet empirically effective approach. Section 6 provides empirical evidence demonstrating the efficacy and robustness of this metric against alternatives.

**Complete Algorithm**  The above outline the learning process of a single program (visualized in Figure 2). To enhance the full search pipeline, we integrate beam search (Pryzant et al., 2023) to avoid getting trapped in local minima. Specifically, each iteration of the learning algorithm maintains and expands $B$ trees, where $B$ represents the beam size. Algorithm 2 in Appendix A.7 summarizes the entire process.

## 5 EXPERIMENTAL RESULTS

We adopt a comprehensive approach to extensively evaluate the effectiveness of LSPs against various baselines under different settings. Our empirical study is designed to validate the benefits of LSPs over alternative methods by addressing the following research questions:

- *Q1: How does LSP compare against traditional NSPs in expressiveness and interpretability?* We assess this through both quantitative and qualitative evaluations (human studies). (Section 5.2)
- *Q2: Does LSP generalize better than traditional NSPs under domain shifts?* This question is explored in detail in (Section 5.2).
- *Q3: Is the incorporation of explicit structures beneficial to LSPs?* We compare the structured LSP with vanilla prompt optimization, which exemplifies a special case of LSP with a single LLM module. (Section 5.3)
- *Q4: How effective are different LLMs in implementing LSP?* We conduct cross-model experiments to evaluate the performance of various LLMs as the computational backbone for learning and inference in LSP. (Section A.5.1)

### 5.1 GENERAL SETTINGS

**Evaluation**  For language tasks, we test popular LLMs, including GPT-3.5 (`turbo-1104`) (Ouyang et al., 2022), GPT-4 (`1106-preview`) (Achiam et al., 2023), and Gemini-M (`1.0-pro`) (Team et al., 2023). For vision tasks, GPT-4V (`1106-vision-preview`) and Gemini-Vision (`1.5-flash`) are utilized. All experiments are repeated with 3 seeds.

Table 2: **Classification accuracy comparison with XAI methods on IL-Bench-Vision.** Here, all numbers for LSP are obtained with Gemini-Vision as the learner and inference LLM, except for LSP (GPT-4V) which uses the larger GPT-4V as the learner; Decision Tree, operating directly on pixel data, lacks human interpretability. Key findings include: (1) Our method outperforms XAI baselines with an average accuracy of 95.67%, which is over 10% higher than the nearest competitor. (2) The program generated by LSP also demonstrates superior transferability to human raters, as they are able to reproduce the predictions following rules learned by LSP.

| IL-Bench-Vision | | | Palworld | | | | | | |
|---|---|---|---|---|---|---|---|---|---|
| MLLM | Method | Mean | Fire-1 | Fire-2 | Dragon-1 | Dragon-2 | Electric-1 | Electric-2 | Water-1 |
| Gemini-M | Decision Tree (Chen & Guestrin, 2016) | 68.20 | 91.11 ± 12.57 | 32.00 ± 9.80 | 68.33 ± 10.27 | 48.33 ± 20.95 | 82.67 ± 6.80 | 65.33 ± 13.60 | 66.67 ± 8.50 |
| | ProtoTree (Nauta et al., 2021b) | 84.33 | **100.00 ± 0.00** | 62.67 ± 12.36 | 98.33 ± 2.36 | 85.00 ± 4.08 | **100.00 ± 0.00** | 82.67 ± 9.98 | 61.67 ± 25.93 |
| | LSP | **96.83** | 93.33 ± 0.00 | **92.00 ± 0.00** | **100.00 ± 0.00** | **100.00 ± 0.00** | **100.00 ± 0.00** | 95.00 ± 5.00 | **97.50 ± 2.50** |
| | LSP (GPT-4V) | 95.67 | 96.67 ± 3.33 | 90.00 ± 6.00 | 90.00 ± 10.00 | 97.50 ± 2.50 | **100.00 ± 0.00** | **98.00 ± 2.00** | **97.50 ± 2.50** |
| Human Rater | ProtoTree (Nauta et al., 2021b) | 72.74 | 83.33 ± 16.67 | 50.0 ± 10.0 | **100.0 ± 0.0** | 75.0 ± 0.0 | 83.33 ± 16.67 | 80.0 ± 0.0 | 37.5 ± 12.5 |
| | LSP (GPT-4V) | **90.36** | **100.00 ± 0.00** | **70.00 ± 10.00** | **100.00 ± 0.00** | 87.5 ± 12.5 | **100.00 ± 0.00** | **100.00 ± 0.00** | 75.00 ± 25.00 |

**Implementation details of LSP**   Our default model of choice is GPT-3.5 for language tasks and Gemini-Vision for vision tasks for cost efficiency, but also examine cross-(M)LLM performance in Appendix. All LLM modules are initialized with an empty instruction "none". More detailed hyperparameters can be found in Appendix A.8, which is kept fixed throughout the experiments.

## 5.2 COMPARISON WITH TRADITIONAL INTERPRETABLE LEARNING METHODS

We compare LSP with two established models - ProtoTree (Nauta et al., 2021b) and Decision Tree (Chen & Guestrin, 2016) - both organize prediction process in tree-structured formats. Among existing NSP methods, the closest to ours is ProtoTree - a highly interpretable NSP that learns a discrete binary tree end-to-end, where each node stores an image patch ("prototype") and the edges determine whether the prototype exists within the query image. Note that ProtoTree does not rely on an explicit DSL - we could not compare with methods based on explicit DSL since they require domain experts to design those operation, while our goal is to automate the whole process. Since ProtoTree only implements image tasks, this comparison also focus on the vision tasks in IL-Bench.

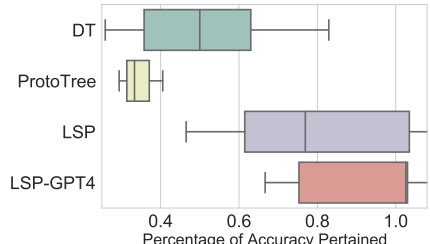

Figure 3: **Accuracy retention rate on Out-Of-Distribution variants of IL-Bench-Vision.** We compute the ratio of test accuracy evaluated on OOD datasets to the original test accuracy. LSP shows strong transferability to OOD data. Notably, LSP with GPT-4V as the learner retains 90-100% of the original test accuracy.

**Expressiveness**   The expressiveness of the learned programs is evaluated in Table 2. LSP (GPT4) outperforms ProtoTree with an average accuracy of 95.67% - over 10% gain. Considering that GPT/Gemini has never observed the images in our datasets before (curated after their knowledge cutoff), this result suggests LSP is capable of formulating effective predictive rules from previously unseen examples.

**Interpretability**   We measure the interpretability of LSPs and NSPs by having human raters make predictions based on visualizations of the learned programs (See Appendix for evaluation protocols). This process essentially "transfers" knowledge from models back to human. Notably, many XAI methods fall short of achieving this level of interpretability, with ProtoTree being a rare exception. As summarized in Table 2, the program generated by LSP also demonstrates stronger transferability to human raters, as they are able to largely reproduce the predictions following rules learned by LSP.

**Generalization under Domain Shift**   In contrast to traditional NSP models that rely on parametric memory, LSP utilizes language instructions to encode knowledge. This strategy significantly enhances robustness against variations in visual attributes (domain shifts). To verify this advantage, we examine the transferability of the learned programs to Out-of-Distribution (OOD) data, constructed using GPT-4V (See Appendix for details) As shown in Figure 3, LSP demonstrates exceptional resilience to domain shifts, compared with ProtoTree.

## 5.3 COMPARISON WITH PROMPT OPTIMIZATION METHODS

Since there exists a variety of PO method that primarily differ in the search algorithm, we select one most representative method from each major category: Monte Carlo sampling (APE) (Zhou et al.,

**Table 3: Classification accuracy comparison with Prompt Optimization methods on IL-Bench-Language.** Key findings: (1) LSP achieves $\sim 6\%$ accuracy gain over the second best method, PromptAgent, with comparable search and inference costs. (2) Across synthetic Decision Tree datasets categorized by increasing complexity of oracle decision rules (Easy, Medium, Hard), LSP consistently outperforms other methods in maintaining high accuracy levels, demonstrating its superior ability to reverse-engineer complex rules from observed data.

| Text Benchmark | | | | Symbolic | | | Caption | | | | | |
|---|---|---|---|---|---|---|---|---|---|---|---|---|
| Method | Mean Acc | Search Cost | Infer Cost | DT-Easy | DT-Medium | DT-Hard | Waxwing | Waterthrush | Jaeger | Albatross | Blackbird | Swallow |
| APE (Zhou et al., 2022) | 67.42 | 270.60s | 0.11s | $100.00 \pm 0.00$ | $85.00 \pm 4.42$ | $75.67 \pm 4.52$ | $50.00 \pm 2.72$ | $45.00 \pm 3.60$ | $66.11 \pm 2.83$ | $48.89 \pm 3.14$ | $80.00 \pm 3.12$ | $56.11 \pm 2.39$ |
| OPRO (Yang et al., 2023a) | 55.48 | 257.86s | 0.14s | $50.00 \pm 1.08$ | $50.17 \pm 3.06$ | $30.33 \pm 2.62$ | $57.22 \pm 2.08$ | $57.22 \pm 4.16$ | $76.67 \pm 4.71$ | $40.37 \pm 3.43$ | $78.06 \pm 2.83$ | $55.28 \pm 1.04$ |
| APO (Pryzant et al., 2023) | 70.67 | 270.85s | 0.08s | $100.00 \pm 0.00$ | $96.67 \pm 4.71$ | $77.83 \pm 11.90$ | $56.11 \pm 4.78$ | $48.89 \pm 4.16$ | $70.00 \pm 5.93$ | $54.07 \pm 9.70$ | $74.17 \pm 2.97$ | $58.33 \pm 1.36$ |
| TreePrompt[†](Singh et al., 2023) | 65.64 | 301.52s | 0.34s | $100.00 \pm 0.00$ | $83.50 \pm 6.68$ | $57.83 \pm 5.89$ | $55.00 \pm 7.20$ | $53.33 \pm 4.91$ | $73.89 \pm 1.57$ | $47.78 \pm 1.57$ | $65.56 \pm 0.39$ | $53.89 \pm 2.08$ |
| PromptAgent (Wang et al., 2023) | 72.40 | 220.95s | 0.11s | $97.67 \pm 3.30$ | $88.50 \pm 8.44$ | $64.33 \pm 20.27$ | $60.56 \pm 4.78$ | $56.67 \pm 6.24$ | $75.00 \pm 3.60$ | $\mathbf{74.44 \pm 6.54}$ | $74.17 \pm 1.36$ | $57.22 \pm 0.79$ |
| LSP (Ours) | $\mathbf{78.53}$ | 232.54 | 0.13s | $99.83 \pm 0.24$ | $\mathbf{99.00 \pm 0.82}$ | $\mathbf{96.83 \pm 0.85}$ | $\mathbf{65.83 \pm 4.17}$ | $\mathbf{62.50 \pm 0.83}$ | $\mathbf{80.00 \pm 1.67}$ | $61.11 \pm 1.11$ | $\mathbf{78.75 \pm 0.42}$ | $\mathbf{62.92 \pm 0.42}$ |

† TreePrompt is a pre-LLM era prompt optimization methods. We adapt this method to support LLMs. See Appendix A.8 for more details.

**Table 4: Classification accuracy comparison with Prompt Optimization methods on IL-Bench-Vision.** LSP achieves an average accuracy of 96.83%, which is $\sim 20\%$ higher than the 2nd best method (APO).

| Vision Benchmark | | | | Palworld | | | | |
|---|---|---|---|---|---|---|---|---|
| Method | Mean | Fire-1 | Fire-2 | Dragon-1 | Dragon-2 | Electric-1 | Electric-2 | Water-1 |
| APE (Zhou et al., 2022) | 47.45 | $60.00 \pm 0.00$ | $38.00 \pm 18.00$ | $43.33 \pm 3.33$ | $42.50 \pm 7.50$ | $53.33 \pm 0.00$ | $25.00 \pm 15.00$ | $70.00 \pm 15.00$ |
| OPRO (Yang et al., 2023a) | 28.09 | $13.33 \pm 0.00$ | $20.00 \pm 0.00$ | $30.00 \pm 10.00$ | $25.00 \pm 0.00$ | $53.33 \pm 20.00$ | $25.00 \pm 0.00$ | $30.00 \pm 0.00$ |
| APO (Pryzant et al., 2023) | 76.38 | $70.00 \pm 16.67$ | $58.00 \pm 10.00$ | $96.67 \pm 3.33$ | $77.50 \pm 2.50$ | $90.00 \pm 10.00$ | $67.50 \pm 2.50$ | $75.00 \pm 5.00$ |
| TreePrompt (Singh et al., 2023) | 67.20 | $60.00 \pm 0.00$ | $50.00 \pm 6.00$ | $93.33 \pm 6.67$ | $77.50 \pm 2.50$ | $53.33 \pm 0.00$ | $65.00 \pm 20.00$ | $70.00 \pm 0.00$ |
| PromptAgent (Wang et al., 2023) | 66.33 | $53.33 \pm 40.00$ | $56.00 \pm 4.00$ | $96.67 \pm 3.33$ | $72.50 \pm 17.50$ | $63.33 \pm 16.67$ | $55.00 \pm 20.00$ | $67.50 \pm 27.50$ |
| LSP (Ours) | $\mathbf{96.83}$ | $\mathbf{93.33 \pm 0.00}$ | $\mathbf{92.00 \pm 0.00}$ | $\mathbf{100.00 \pm 0.00}$ | $\mathbf{100.00 \pm 0.00}$ | $\mathbf{100.00 \pm 0.00}$ | $\mathbf{95.00 \pm 5.00}$ | $\mathbf{97.50 \pm 2.50}$ |

2022), evolutionary search (ORPO) (Yang et al., 2023a), beam search (APO) (Pryzant et al., 2023), and tree search (PromptAgent) (Wang et al., 2023). We also adapt TreePrompt (Singh et al., 2023) - a pre-LLM method that fits a classic decision tree to a set of pre-defined prompts - to LLMs. Since the main bottleneck for PO methods is the candidate evaluation, we follow existing works and set the same maximum number of candidate proposals for all methods (100 candidates).

**Results** The empirical results indicate that incorporating explicit structures significantly enhances performance of the programs on predictive tasks: LSP consistently outperforms all vanilla prompt optimization methods, with a considerable margin of $20.09\%$ and $4.89\%$ over the 2nd best methods on vision and language tasks respectively. The advantages of integrating structured learning are twofold: (1) It simplifies the learning process: LSP benefits from a divide-and-conquer approach where each LLM-module node focuses solely on extracting predictive rules for a specific subset of the data. (2) It streamlines the inference process: We observe that LLMs tend to exhibit hallucination as the complexity of the instructions increases (e.g., multiple conditional clauses. In contrast, LSP mitigates this issue by ensuring that each LLM module contains simpler, more manageable instructions.

**Search cost analysis** A key advantage of the structured prediction approach in LSP is that theoretically, it can reduce inference costs when executing oracle decision rules. This efficiency arises because, during prediction, only a small subset of branches is executed for a given test input, and the prompt on each branch is also much simpler due to divide-and-conquer. Consequently, we observe empirically that LSP's search and inference costs are comparable to those of various prompt optimization baselines (Table 3). For a more detailed analysis, please refer to Appendix A.4.

# 6 ABLATION STUDY

**Convergence of LLM-Symbolic Program LSP** LSP organizes instructions into a tree-based structure. Such divide-and-conquer strategy simplifies the learning process. To verify this, we also plot the training trajectories for LSP across various tasks. The training trajectory indicates the how fast a model fits the observed examples. As Figure 5 demonstrates, LSP not only converges faster but also achieves higher final accuracy compared to models that use unstructured prompting techniques.

**Different node scoring functions** Table 5 summarizes the performance of LSP using three different node scoring functions: (1). Error count. (2). Prediction accuracy. (3). Random scoring. The results suggest that error count performs more consistently across different tasks.

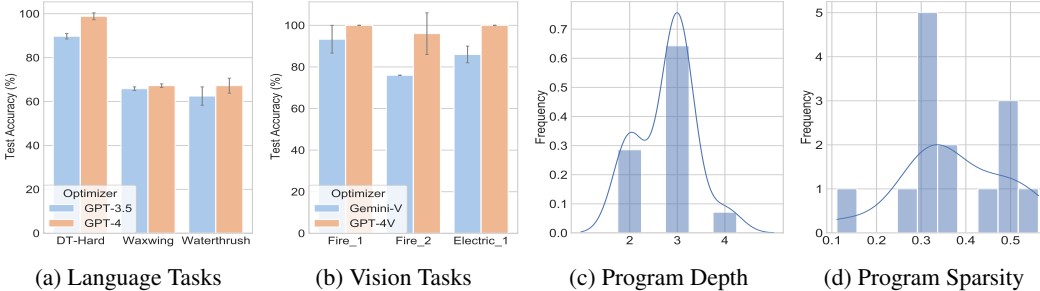

| (a) Language Tasks | (b) Vision Tasks | (c) Program Depth | (d) Program Sparsity |

Figure 4: **(a, b): Stronger LLMs as better LSP learners.** In these experiments, we keep the inference LLM fixed (GPT-3.5 for text and Gemini-V for images) while swapping the learner LLM with GPT-4. With its larger parameter count, GPT-4 consistently achieves better performance in learning LSPs. **(c, d): Statistics of discovered programs.** Averaged from the IL-Bench-Language tasks, the resulting LSPs are generally shallow and sparse, indicating that the final prediction can be reached within only a few steps.

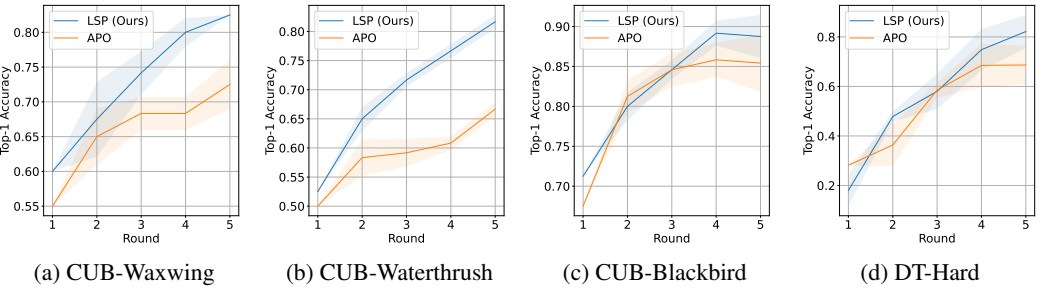

| (a) CUB-Waxwing | (b) CUB-Waterthrush | (c) CUB-Blackbird | (d) DT-Hard |

Figure 5: **Convergence of different algorithms across time**. We plot the trajectory of training accuracy against the number of optimization rounds. The API model is GPT-3.5. (1). LSP converges substantially faster than vanilla prompting; (2). The search process does not introduce extra variances.

**Robustness to meta-prompts** LLM's behavior is highly sensitive to prompt formulation, where even minor variations in prompts might lead to significantly different outcomes. To assess the robustness of LSP's performance against variations in the meta-prompt - the prompt used by the learner LLM to generate rules - we conducted experiments with three alternative prompts. These prompts were paraphrased versions generated by distinct LLMs (visualized in Appendix A.5). The results, presented in Table 5, indicate that LSP's performance remains consistent across all meta-prompt variants, demonstrating robustness to prompt formulation.

**Complexity of discovered programs** We found that the complexity of programs developed by LSP is fairly manageable: Most programs can reach a final prediction within just three steps, as illustrated in Figure 4c, and the tree structures tend to be sparse, as shown in Figure 4d. These observations confirm that although theoretical maximum tree expansion could grow exponentially with depth, in practice, LSPs operate effectively without requiring overly complex structures.

Table 5: **Comparison of Different Node Scoring Functions** on three tasks from IL-Bench-Language. Despite its simplicity, error count achieves more consistent performance compared to alternative metrics.

| Node Scoring | DT-Hard | Waxwing | Waterthrush |
|---|---|---|---|
| Random | $70.50 \pm 11.01$ | $62.22 \pm 4.78$ | $61.67 \pm 1.36$ |
| Accuracy | $80.33 \pm 18.27$ | $\mathbf{66.11 \pm 7.86}$ | $54.44 \pm 0.70$ |
| Error Count (LSP) | $\mathbf{96.83 \pm 0.85}$ | $65.83 \pm 4.17$ | $\mathbf{62.50 \pm 0.83}$ |

| Meta Prompt | DT-Hard | Waxwing | Waterthrush |
|---|---|---|---|
| Paraphrase-1 | $97.50 \pm 2.12$ | $65.00 \pm 4.91$ | $\mathbf{66.11 \pm 3.14}$ |
| Paraphrase-2 | $98.50 \pm 0.71$ | $61.67 \pm 2.36$ | $62.22 \pm 3.93$ |
| Paraphrase-3 | $\mathbf{99.33 \pm 0.62}$ | $62.78 \pm 2.83$ | $63.89 \pm 0.79$ |
| Original (LSP) | $96.83 \pm 0.85$ | $\mathbf{65.83 \pm 4.17}$ | $62.50 \pm 0.83$ |

## 7 CONCLUSION

This work aims at revitalizing the concept of Neuro-Symbolic Programming in the era of Large Language Models. We demonstrate that pretrained LLMs can implement powerful symbolic programs that are expressive, interpretable, and easy to train. Additionally, we introduce the Instruction Learning Benchmark (IL-Benchmark), which consists of a suite of vision and language datasets designed to evaluate instruction learning algorithms. We hope that our proposed framework will inspire new developments in interpretable learning methods during the LLM era. We regard our study as an initial step in the research on LLM-Symbolic Programs. Accordingly, we acknowledge the limitations of the current method in Appendix Section A.11.

ACKNOWLEDGMENT

This work is partially supported by NSF 2048280 and 2331966.

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

# A  SUPPLEMENTAL MATERIAL

**Organization**    The appendix file is organized as follows:

## A.1  MORE DETAILS ON RELATED WORK

**Interpretable machine learning**    Although neural networks are immensely expressive, they provide no insights into its internal decision making mechanism. In the quest of making model predictions interpretable, research has broadly categorized methods into two main types: post-hoc and intrinsic. Post-hoc methods provide insights into how a pretrained model behaves, usually by highlighting important features used for decision making (Zintgraf et al., 2017; Petsiuk et al., 2018; Dabkowski & Gal, 2017; Shrikumar et al., 2017; Sundararajan et al., 2017; Ancona et al., 2017) or provide counterfactual explanations (Dhurandhar et al., 2018; Hendricks et al., 2018; van der Waa et al., 2018; Goyal et al., 2019; Hsieh et al., 2021). Beyond attribution in the feature space, some methods can also be generalized to the space of higher level concepts (Kim et al., 2018; Bai et al., 2023). However, all these methods aim to highlight important features while not being able to recover the entire decision making process of neural networks.

On the other hand, intrinsic methods integrate interpretability directly into the model's architecture, making them naturally interpretable by design. Traditional Methods include Decision Trees (Chen & Guestrin, 2016) and Generalized Additive Models (GAMs) (Hastie & Tibshirani, 1990) offer strong interpretability, yet often not expressive enough. Concept bottleneck model adds a hidden layer in neural network, where neurons represent some predefined concepts to gain interpretability (Koh et al., 2020; Losch et al., 2019; Yuksekgonul et al., 2022; Oikarinen et al., 2023). While this approach facilitates attribution of concepts, it does not provide a comprehensive decision rule, and the concepts need to be predefined by human experts. In contrast, LSP directly learns all interpretable modules (LLM prompts) from data without relying on human prior knowledge. Furthermore, LSP fully reveals its decision process through learned prompts and program structure, while concept-based methods only partially expose the decision process. Neurosymbolic Programming (NSP) (Chaudhuri et al., 2021; Shah et al., 2020; Cui & Zhu, 2021; Nauta et al., 2021b) represents an innovative blend, combining deep learning's data handling capabilities with symbolic reasoning to foster both performance and transparency. Despite early promises, NSP suffers from an inherit trade-off between expressiveness (more NN modules) and interpretability (more symbolic modules). Moreover, they are often expensive to train due to co-optimization of program architecture and parameters of the NN modules (Shah et al., 2020; Cui & Zhu, 2021).

**Prompt Optimization**    The essence of utilizing a generative language model lies in crafting effective prompts. Recent advancements have aimed to automate this process, reducing the need for human effort through prompt optimization (Shin et al., 2020; Zhou et al., 2022). While pioneering efforts were mainly directed towards various discrete optimization algorithms (Shin et al., 2020; Deng et al., 2022; Zhang et al., 2022; Wang et al., 2024b), it has been noted that advanced LLMs can revise prompts similarly to human engineers (Zhou et al., 2022; Pryzant et al., 2023; Wang et al., 2024a).

Since these initial efforts, a significant body of research has emerged, exploring various search algorithms including Monte Carlo Sampling (Zhou et al., 2022), beam search (Pryzant et al., 2023), evolutionary search (Yang et al., 2023a; Fernando et al., 2023; Xu et al., 2022; Guo et al., 2023; Hsieh et al., 2023), and tree search (Wang et al., 2023). However, existing methods often treat the prompt as a single entity without explicit structure. From this perspective, prompt optimization methods can be seen as simplified instances of LSPs, where the program consists solely of one LLM module. While this simplification has shown promising results, as task complexity increases, the explicit structuring within LSPs allows them to encode knowledge from data. This provides substantial advantages over conventional prompt optimization methods. The only exception is TreePrompt (Singh et al., 2023), developed before the LLM era. TreePrompt first pre-generates a set of prompts as attributes and fits a decision tree on top of them. On the other hand, LSP aims at establishing a principled hybrid between LLMs and NeuroSymbolic Programming, which substantially differs from traditional decision tree algorithms in program structure search, module definition, module learning method, and extendability. Concretely, LSP uses progressive tree search algorithm to search for program structures; Moreover, all LLM modules are fully optimized by LLMs using the proposed rule learning prompting method; The LLM module on each node are trained to fit subset of data assigned to it instead of capturing the full data distribution, making the learning task much simpler. Similar to NSP, LSP framework also enjoys great extendability, allowing us to seamlessly incorporate extra modules (either learned or manually defined) to the search space to include more complex and tailored programs for new tasks. Empirical results also suggest that LSP achieves substantial gain over previous prompt optimization method.

**Augmenting LLMs with Neural-Symbolic Solvers**   Symbolic AI encompasses a diverse set of methods and tools suitable for various applications. Although prior work has explored combining symbolic approaches with LLMs, these efforts target distinct tasks compared to LSP (Dong et al., 2023; Fang et al., 2024; Yang et al., 2023b). For instance, Dong et al. (2023) focuses on enhancing LLMs' story comprehension ability by converting storylines into code, while Fang et al. (2024); Yang et al. (2023b) augment LLMs with external symbolic solvers to improve accuracy. These approaches are not applicable to the Intepretable Learning task that our work addresses.

## A.2 MORE DETAILS ON IL-BENCH

### A.2.1 DATA CURATION AND STATISTICS

**Symbolic tasks**   For symbolic tasks, we use $x_i{}_{i=1}^M$ to represent input variables, with values denoted by $A_j, B_j, C_j, \ldots$. The label for each data point takes values from $0, 1, 2, \ldots, N-1$. Inspired by the natural alignment of many decision-making processes with tree structures, we use synthetic decision trees to generate labels for each data point.

Each level of the decision tree processes one variable, and leaf nodes are assigned so that labels are evenly distributed. The dataset is generated by randomly sampling a value for each variable and then passing the resulting example through the decision tree. The parameters $M$ and $N$ are predefined to control task difficulty: more variables increase the complexity of the underlying rules, making the task more challenging for the model. This setup allows for automatic generation of symbolic tasks that can be extended to arbitrarily high levels of difficulty.

**Language tasks**   For the initial version of IL-Bench, we primarily use the CUB dataset (Wah et al., 2011) to construct text classification tasks, though the curation method presented here can be readily applied to convert any visual classification dataset (e.g., Stanford Cars, Dog Breeds, Food Items (Maji et al., 2013)), which we plan to add in future releases. CUB is a fine-grained visual classification dataset comprising visually similar bird subspecies, making it widely used in pre-LLM-era interpretability research.

To convert this dataset into text classification tasks, we use GPT-4 as the captioner. Since an image contains far richer information compared to a text modality, captioning images individually risks missing fine-grained details that are crucial for distinguishing between bird subspecies, which could render the task ill-defined. To address this, we generate contrastive captions: for each target image, we sample images from other classes as contrastive examples. This contrastive approach is applied for every class, and all resulting captions are concatenated to form the input for the new text classification dataset. To avoid information leakage through label names, class names (e.g., `North_American_Waterthrush`) are replaced with symbols (e.g., `class_1`).

Empirically, we confirmed that the curated datasets are not solvable in a zero-shot setting: all tested LLMs in our experiments could not outperform random guessing without learning the underlying rules.

**Vision tasks**   To curate images that are unfamiliar to the MLLMs, we use a regional Pokémon-style video game called "Palworld," which contains approximately 150 creatures ("Pals") of different types (e.g., water, fire, electric). To make the task challenging, we group visually similar Pals into the same dataset. Since these visually similar Pals often belong to the same type, we name each dataset according to the type (e.g., `fire_1`). All images are collected via screenshots of publicly available in-game footage on YouTube. Similar to the language tasks, Pal names are replaced with symbols to prevent information leakage.

### A.2.2   Task descriptions and examples

Table 8 provides an overview of each task in IL-Bench, including task name, input modality, descriptions, and example data points.

### A.3   Qualitative analysis of discoverd programs

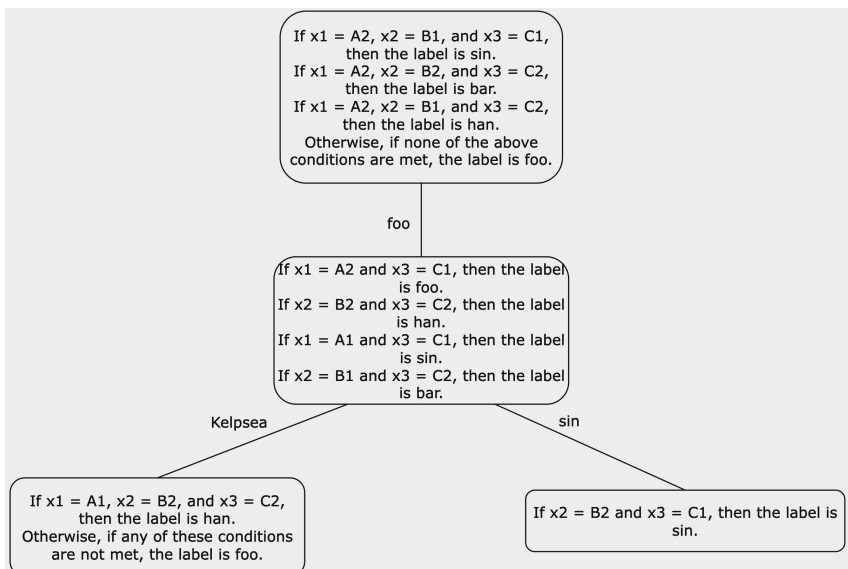

Figure 6: **Example program discovered by LSP on DT-Hard task.**

In this section, we provide qualitative analysis of the discovered programs. We use programs discovered from DT-Hard task as illustrating example, as knowing the oracle rules for this task allows us to precisely identify the reasons for both success and failure.

The data for the DT-Hard task are generated using the following rules:

- `Label = foo when x1=A1, x2=B1, x3=C1 or x1=A2, x2=B2, x3=C1`
- `Label = bar when x1=A1, x2=B1, x3=C2 or x1=A2, x2=B2, x3=C2`
- `Label = sin when x1=A1, x2=B2, x3=C1 or x1=A2, x2=B1, x3=C1`
- `Label = han when x1=A1, x2=B2, x3=C2 or x1=A2, x2=B1, x3=C2`

Figure 6 visualizes an example program discovered by LSP, which achieves 96% test accuracy. Here, nodes are LLM modules with rules, and edges denote the prediction from the parent node. If the rule on a specific node cannot cover a test query, it will simply return its parent's prediction. By examining the program, we can observe that it learns to "divide-and-conquer" a test query: Take the rules at the root node as an example, it first summarizes a few rules for label sin, bar and han, but decide to classify every other situations as foo; This is clearly not accurate, so its child node further refines the rules. Let us use the data point `"x1=A1, x2=B2, x3=C1"` as an example. At the root node, the rule states `"Otherwise, the label is foo"`, which sends this example

to the child node. At this child node, the rule becomes `"if x1=A1 and x3=C1, label as sin"`, which sends this example to the left child node. At this leaf node, the rule is `"if x2=B2, x3=C1, the the label is sin"`, resulting in the final prediction of `sin`, which is correct.

From this representative example, the following observations can be made:

- The root node initially misclassifies the example as `"sin"`, demonstrating that current LLMs can still make errors when generating predictive rules.
- However, this error is corrected by the child node, resulting in an accurate final prediction.
- The rules at each node need not be complete, as child nodes are responsible for correctly predicting the subset of data assigned to them.
- There exists redundancy between the rules at different nodes, this suggests that the learned program could be further simplified using post-hoc algorithms.

## A.4 DETAILED COMPLEXITY ANALYSIS OF LSP

LSP follows a multi-step decision-making process, akin to a decision tree. While this might initially suggest an increase in inference time, in-depth complexity analysis demonstrates that LSP actually improves inference efficiency.

**Inference cost depends on total token count, not number of prompts** Assuming network speed is not a bottleneck, the inference cost is primarily determined by the total token count rather than the number of prompts. Although LSP necessitates multiple LLM calls for a final prediction, the individual prompts are significantly simpler and shorter, due to the divide-and-conquer strategy. While LSP requires multiple LLM calls to reach a final prediction, each prompt is significantly simpler and shorter due to LSP's divide-and-conquer strategy.

**Tree structure of LSP reduces theoretical inference cost** Consider an oracle rule represented with $N$ tokens. If represented in a traditional prompt, the inference LLM must process $\mathcal{O}(N)$ tokens. By contrast, using LSP's complete binary tree structure, the LLM processes only $\mathcal{O}(N/\log D)$ tokens per test query, where $D$ represents the program depth (with some minor template overhead in practice). This is because only one path in the LSP tree are executed for a given test input, thereby substantially reduces the inference cost of oracle rules.

**Oracle rules are naturally complex and lengthy** The oracle rules underlying many datasets, particularly those from IL-Bench, tend to be inherently complex. Such rules are often composed of simpler sub-rules, resulting in longer token sequences. As the complexity of an oracle rule increases, the minimal description length (measured by token count) also grows, naturally raising the inference cost. Importantly, no token limit was imposed on any of the baselines, allowing them to introduce more rules if beneficial. However, unstructured learning methods often produce relatively simple prompts that perform worse. In practice, LSP only uses comparable or slightly more tokens than previous SOTA, while is substantially more accurate in captures the complex oracle decision rules.

## A.5 ADDITIONAL ABLATION EXPERIMENTS

### A.5.1 USING DIFFERENT LLMS TO IMPLEMENT LSPS

The role of LLMs in LSPs is twofold: they serve both as the inference and learning engine of the LLM-modules in the grammar. The learning engine is responsible for summarizing and organizing patterns from observed data samples into clear predictive rules, whereas the inference engine follows the learned program to make predictions on test examples. Natural questions arise: (1). how effective are different LLMs at optimizing LSPs? (2). Is the learned programs interpretable to different LLMs?

**LLM as LSP learner** We replace the learning engine used in optimizing LSP with various LLMs - GPT-3.5, Gemini, and GPT-4 - while keeping all other settings consistent with the main experiment. As shown in Figure 4, GPT-4 consistently outperforms other LLMs on both text and vision tasks, while Gemini and GPT-3.5 show similar performance with each other. This reflects their respective capabilities. For specific examples of instructions generated by different LLM optimizers, please see the Appendix.

Table 6: **Transferring LSPs learned from one LLM to another.** The learned LSPs are generally interpretable across various LLMs. However, larger LLMs (e.g., GPT-4) demonstrate a slightly higher consistency in understanding LSPs learned by other LLMs.

| Source Model | Task | Evaluator | | |
|---|---|---|---|---|
| | | GPT3.5 | Gemini-M | GPT4 |
| GPT3.5 | DT-Hard | $89.75 \pm 1.25$ | $72.67 \pm 6.91$ | $87.50 \pm 1.22$ |
| | Waxwing | $65.83 \pm 4.17$ | $52.22 \pm 1.57$ | $56.67 \pm 3.60$ |
| | Waterthrush | $62.50 \pm 0.83$ | $64.44 \pm 0.79$ | $59.44 \pm 3.93$ |
| Gemini-M | DT-Hard | $75.50 \pm 2.04$ | $80.83 \pm 1.03$ | $79.17 \pm 11.45$ |
| | Waxwing | $52.78 \pm 3.42$ | $58.33 \pm 4.91$ | $61.11 \pm 10.57$ |
| | Waterthrush | $50.56 \pm 4.16$ | $54.44 \pm 5.50$ | $52.22 \pm 0.79$ |
| GPT4 | DT-Hard | $74.50 \pm 9.35$ | $57.67 \pm 3.01$ | $99.50 \pm 0.00$ |
| | Waxwing | $59.44 \pm 5.15$ | $62.22 \pm 7.49$ | $63.33 \pm 4.91$ |
| | Waterthrush | $66.67 \pm 6.80$ | $68.33 \pm 2.72$ | $62.78 \pm 9.06$ |

**LLM as LSP interpreter**    We then test if LSPs created by one LLM could be interpreted by other LLMs. Table 6 summarizes the performance. The results suggest that LSPs are interpretable across a diverse range of inference models; Larger and stronger LLMs (e.g. GPT-4) demonstrates a slight more consistent ability in interpreting LSPs, which aligns their superior instruction-following capacities.

## A.6    DIFFERENT PARAPHRASING OF THE META-PROMPT

Here, we visualize the different paraphrased version of the meta-prompt used in Table 5.

| Version | Prompt |
|---|---|
| Paraphrasing-1 | Begin by outlining the patterns visible in these examples; Next, formulate one well-defined rule that successfully predicts the labels for these examples using these patterns. |
| Paraphrasing-2 | Start by identifying and explaining the patterns found in these examples; Then, propose one robust rule that can accurately predict the labels based on the identified patterns. |
| Paraphrasing-3 | Start by identifying the patterns in these examples; then, develop a clear rule that accurately forecasts the labels for these examples based on these patterns. |
| Original | First explain the patterns you observe from the above examples; Then provide 1 high-quality rule that can correctly predict the labels of those examples based on those patterns. |

Table 7: Different variants of the meta-prompt used by the learner LLM when building LSP. The variants are produced by asking different LLMs to paraphrase the original meta-prompt.

## A.7    LEARNING ALGORITHM FOR LSP

The complete pipeline for constructing LSP is summarized in Algorithm 1 and Algorithm 2.

**Remarks**

- Although initially, the complexity of the program expansion might seem exponential to the tree depth, a closer examination reveals otherwise: (1). In practice, the trees are typically sparse, meaning that expanding only a few branches is often sufficient to achieve good performance (Figure 4d). (2). The divide-and-conquer approach ensures that each tree level processes the same amount of data making the evaluation complexity linear to tree depth.
- The above arrangement of the search process does not compromise generality of LSP: For more sophisticated DSL designs, program structure search can be conducted similarly to traditional NSPs, using top-down tree traversal Chaudhuri et al. (2021); Cui & Zhu (2021).

---

**Algorithm 1** `learn_llm_module`: Learning LLM Module by summarizing predictive rules

---

1: **Input:** Proposal size $m$, data sample $\mathcal{B}$, learner LLM $\mathcal{M}_l$
2: Initialize an empty list of LLM modules $\Phi$
3: **for** $i = 1$ **to** $m$ **do**
4:     Randomly sample $b \sim \mathcal{B}$
5:     $\phi_{new} \leftarrow$ `summarize`$(M_l, b)$
6:     $\Phi \leftarrow \Phi \cup \{\phi_{new}\}$
7: **end for**
8: **return** $\Phi$

---

---

**Algorithm 2** Complete pipeline of optimizing LSPs

---

1: **Input:** Dataset $\mathcal{D}$, beam size $d$, number of iterations $T$, inference LLM $\mathcal{M}_i$, learner LLM $\mathcal{M}_l$, expand ratio $K$, proposal size $m$
2: Initialize $p_0$ as an empty program
3: Initialize candidate program set $P = \{p_0\}$
4: **for** $t = 1$ **to** $T$ **do**
5:     **for** each program $p$ in $P$ **do**
6:         *▷ Batch evaluation*
7:         Sample a batch $\mathcal{B} \sim \mathcal{D}$
8:         Evaluate $p$ on $\mathcal{B}$ using $\mathcal{M}_i$
9:         *▷ Selecting the most promising node $n$ to expand*
10:       Assign $\mathcal{B}$ to the leaf nodes of $p$
11:       Identify the most error-prone leaf node $n$ with assigned subset $\mathcal{B}_n$
12:       *▷ Extend program $p$ to $K$ new programs by adding top-$K$ LLM modules to node $n$*
13:       $\Phi \leftarrow$ `learn_llm_module`$(n, \mathcal{B}_n, \mathcal{M}_l, m)$
14:       $\Phi_{topK} \leftarrow$ evaluate and retain top-$K$ $\Phi$ on $\mathcal{B}_n$
15:       $\mathcal{P}_{new} \leftarrow$ extend $p$ by assigning each $\phi \in \Phi_{topK}$ to node $n$ on program $p$.
16:       $\mathcal{P} \leftarrow \mathcal{P} \cup \mathcal{P}_{new}$
17:     **end for**
18:     Evaluate and retain the top-$d$ programs from $\mathcal{P}$ on $\mathcal{D}$
19: **end for**
20: **return** The best program from $P$

---

Table 8: **Overview of Interpretable-Learning Benchmark**. We provide task names, types, summaries, number of labels, and one example data point for each task.

| Task | Type | Summary | Labels | Example |
|---|---|---|---|---|
| DT-Easy | Symbolic | Predict labels based on symbolic inputs. Rules generated by a small decision tree | 2 | "input": "x1=A2; x2=B1", "output": "bar" |
| DT-Medium | Symbolic | Predict labels based on symbolic inputs. Rules generated by a medium decision tree | 2 | "input": "x1=A3; x2=B2", "output": "bar" |
| DT-Hard | Symbolic | Predict labels based on symbolic inputs. Rules generated by a large decision tree | 4 | "input": "x1=A1; x2=B1; x3=C1", "output": "foo" |
| Waxwing | Caption | Classify Waxwing species based on its text description. | 2 | "input": "Tan to light brown head and upper body, black mask across eyes, lighter cream underparts, bright red tips on secondary wing feathers, small black bill, yellow band on tail.", "output": "Cedar Waxwing" |
| Waterthrush | Caption | Classify Waterthrush species based on its text description. | 2 | "input": "Light gray crown, white supercilium, dark eyestripe extending behind eye, olive-brown wings with faint wingbars, white throat, pale underparts, long, slender bill, relatively short tail, orange legs.", "output": "Louisiana Waterthrush" |
| Jaeger | Caption | Classify Jaeger species based on its text description. | 2 | "input": "Light greyish-brown plumage on the underside, distinct narrow white band across the nape, wings with a M-shaped pattern when spread, tail slightly forked but mostly straight across.", "output": "Long tailed Jaeger" |
| Albatross | Caption | Classify Albatross species based on its text description. | 3 | "input": "Dark brown upperparts and paler brown underparts, elongated and narrow wings with a white trailing edge and distinct finger-like tips, hooked beak with a pale base, light-colored head with a dark eye patch and bill, wings held straight in gliding flight, gliding above water surface. Uniform dark brown plumage, long slender wings, distinct white pattern on underwings, white band near the tips of the underwings, pale or white head, dark eye patch.", "output": "Black footed Albatross" |
| Blackbird | Caption | Classify Blackbird species based on its text description. | 4 | "input": "Bright yellow head, black body, sharp conical beak, perched on reed-like vegetation. Bright yellow head, yellow chest, solid black body excluding head and chest, perched on a thin branch. Black body, bright yellow head, sturdy bill, perched on a reed.", "output": "Yellow headed Blackbird" |
| Swallow | Caption | Classify Swallow species based on its text description. | 4 | "input": "Light brown head, pale throat, light brown upperparts, long pointed wings, short tail, white underparts, sitting on wire. Light brown head and upper body, white underparts, sitting on a wire, sky background, short beak, sleek body shape. Brown and white plumage, perched on a wire, stout body, short and thick neck, medium-length tail with a straight edge, compact size, unmarked lighter underparts, darker wings and upperparts.", "output": "Bank Swallow" |
| Fire-1 | Vision | Distinguish visually-similar fire-type pals from Palworld. | 3 | "input": [image] "output:" "Arsox" |
| Fire-2 | Vision | Distinguish visually-similar fire-type pals from Palworld. | 5 | "input": [image] "output:" "Pyrin" |
| Dragon-Blue-1 | Vision | Distinguish visually-similar blue-colored dragon-type pals from Palworld. | 3 | "input": [image] "output:" "Elphidran Aqua" |
| Dragon-Blue-2 | Vision | Distinguish visually-similar blue-colored dragon-type pals from Palworld. | 4 | "input": [image] "output:" "Jetragon" |
| Electric-1 | Vision | Distinguish visually-similar electric-type pals from Palworld. | 3 | "input": [image] "output:" "Grizzbolt" |
| Electric-2 | Vision | Distinguish visually-similar electric-type pals from Palworld. | 4 | "input": [image] "output:" "Univolt" |
| Water-1 | Vision | Distinguish visually-similar water-type pals from Palworld. | 4 | "input": [image] "output:" "Celaray" |

## A.8 IMPLEMENTATION DETAILS

**LSP** Throughout our main experiment, we use an expansion ratio of 4, batch size of 64, a maximum number of four iterations, and a maximum of 8 candidate (LLM module) proposals for each iteration. The settings for beam search follows that of APO, which uses a beam size of 4 and deploys UCBBandits algorithm with a sample size of 32 to speedup the candidate ranking Pryzant et al. (2023). The only exception is that for vision tasks, we use a batch size of 4 for cost reduction. The temperature for all API models are set to their default (0.7).

**Baselines** For all prompt optimization baselines, we set the maximum budget (measured by the number of candidate proposals) to the same number.

- For Decision Tree, we use XGBoost library's standard implementation, which operates on raw pixels.

- For ProtoTree, we directly run the original implementation, but reduce the maximum depth from 9 to 5, as it is faster to train yet achieves better performance on our datasets.

- For TreePrompt, we swap the GPT-2 model used in its implementation with the more capable gpt-3.5-turbo for fair comparison with other more recent baselines.

We align the evaluation our baselines.

## A.9 CONSTRUCTING OUT-OF-DISTRIBUTION DATASET FOR IL-BENCH-VISION TASKS

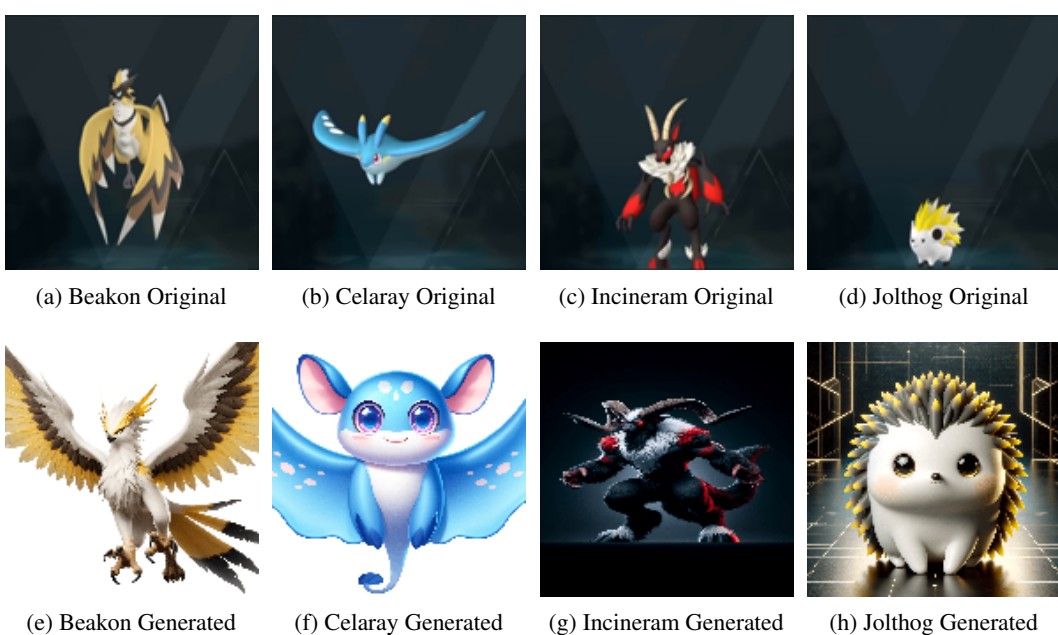

(a) Beakon Original     (b) Celaray Original     (c) Incineram Original     (d) Jolthog Original

(e) Beakon Generated     (f) Celaray Generated     (g) Incineram Generated     (h) Jolthog Generated

Figure 7: **Comparison between original images (top row) and Out-Of-Distribution images (botton row) generated by GPT-4V.** All images are resized to an unified resolution of 128.

Our OOD dataset is constructed by feeding the original image from the training set to GPT-4 (web version), and ask GPT to generate a variant of the input image. The prompt we used is shown below. Figure 7 shows a comparison of some example OOD images generated by GPT-4 with original image.

```
Generate an image variant containing the creature in
the provided image.  keep the key features of this
creature unmodified.  You must show the full body view
of this creature.
```

## A.10 HUMAN EVALUATION PROTOCOL

We conduct user study to access the interpretability of our method and ProtoTree. For both methods, we send (1) the original image datasets and (2) visualizations of the discovered programs to the human raters, and as the human rater to make predictions based on those programs. We then compute the accuracy of their predictions, and report the mean and standard deviations. We select the group of human raters so that they have no background in machine learning research.

## A.11 LIMITATIONS

We acknowledge the following limitations, which merit further exploration in future studies. It is important to note that these limitations pertain to the specific, simplified instantiation of the algorithms used in this preliminary study, rather than to the LSP framework itself:

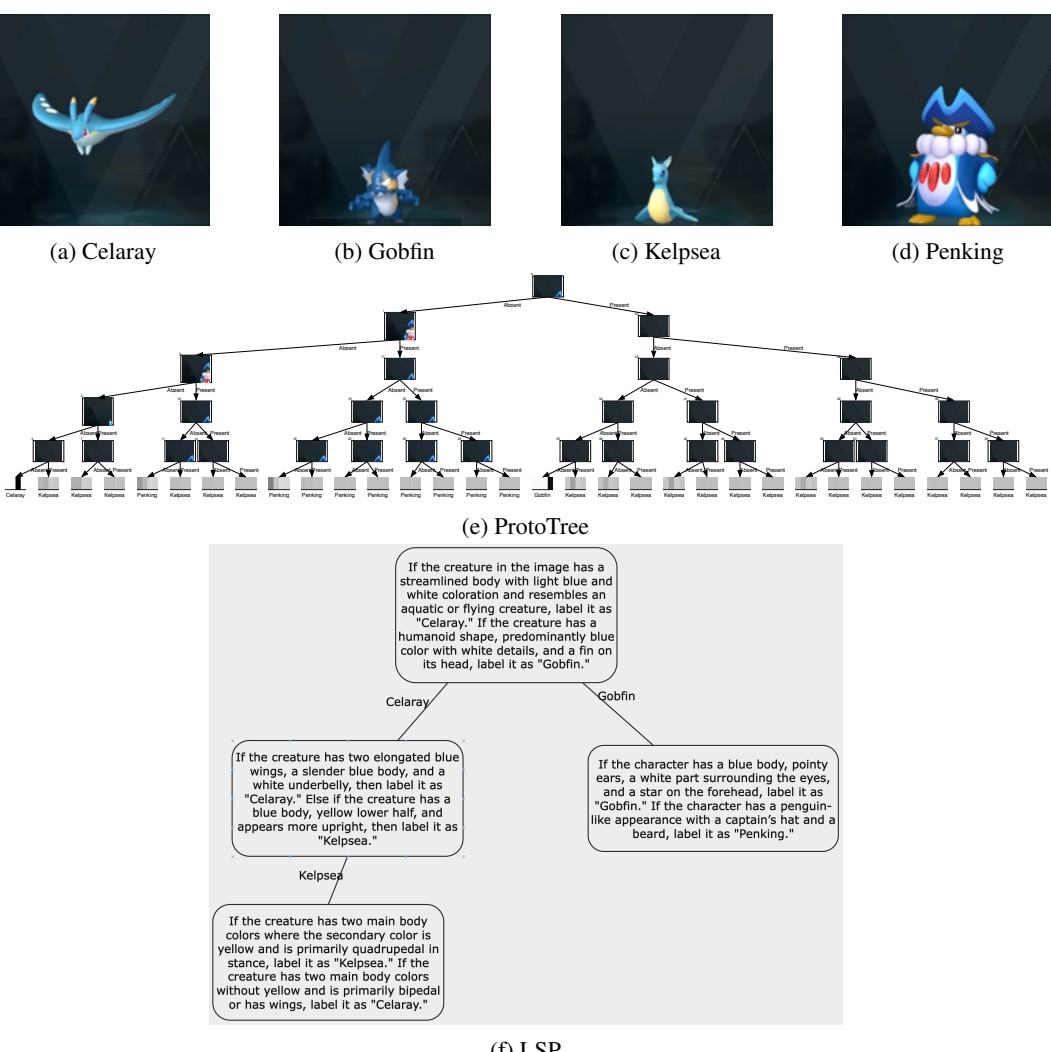

Figure 8: **Example programs discovered by LSP (bottom) and ProtoTree (middle).** While ProtoTree offers some interpretability by displaying prototype image patches to the user, it can be misleading as there is no guarantee that the prototypes are meaningful (e.g. many patches miss the key regions, and there also exists entire branches that overfit to the background). In contrast, the programs discovered by LSP accurately capture the characteristics of the creatures and guide the decision-making process step by step.

- **Domain-Specific Language Design:** A common practice in NSp is to design DSLs suitable for specific tasks. This work presents only a basic example of a DSL designed for predictive tasks. Investigating a variety of DSL designs could enable LSPs to excel across a broader range of applications.
- **Program Complexity:** Our search algorithm prioritizes accuracy without considering the complexity of the resulting programs, potentially leading to redundancies. The complexity of the learned programs could be reduced either through post-processing (akin to code cleaning) or by integrating complexity regularization during the search process.

## A.12 SOCIETAL IMPACT

The development and deployment of interpretable predictive models using Large Language Models (LLMs) have significant societal implications. By enhancing the transparency and interpretability of AI systems, our approach addresses critical concerns related to trust, accountability, and fairness of the decision making process. These improvements are particularly valuable in high-stakes domains such as healthcare, finance, and legal decision-making, where understanding the rationale behind AI decisions is crucial for gaining user trust and ensuring ethical outcomes.

However, as with any AI technology, careful consideration must be given to the potential risks of misuse or unintended consequences. It is essential to continue developing comprehensive guidelines and regulatory frameworks to ensure that the deployment of these models aligns with societal values and ethical standards. By promoting transparency and interpretability, our approach paves the way for more responsible and beneficial integration of AI into society.

## A.13 LICENSE

The open-source code from GitHub used in this paper adheres to various licenses like MIT, Apache 2.0, and GPL, ensuring the code's free use, modification, and distribution under specific conditions. The ChatGPT API from OpenAI and the Gemini API from Google are used in compliance with their respective terms of service, which include usage restrictions, attribution requirements, and provisions for commercial use. By following these licenses and terms, we maintain ethical and legal standards in utilizing both open-source code and proprietary APIs in our research.

