# OpenReview forum: "Large Language Models are Interpretable Learners"
_ICLR.cc/2025/Conference — ICLR 2025 Poster_

### Official Review · Reviewer_ZPkh · 2024-11-01

**Soundness:** 3
**Presentation:** 2
**Contribution:** 3
**Rating:** 8
**Confidence:** 2

**Summary:**

This paper tackles the challenge of balancing expressiveness and interpretability in predictive models for classification and decision-making. The authors propose LLM-based Symbolic Programs (LSPs) that combine Large Language Models with symbolic programs to bridge this gap. LLMs generate interpretable modules that transform raw inputs into natural language concepts, which are then integrated into decision rules by symbolic programs.
Moreover, the paper proposes IL-Bench, a new benchmark for the interpretable learning capabilities of LLMs. In experiments, it is demonstrated that LSPs outperform traditional neurosymbolic programs and prompt tuning methods on the proposed benchmark.

**Strengths:**

The approach to integrating LLMs into DSL-based neuro-symbolic programming approaches is interesting, although the idea itself may be straightforward.
By combining Large Language Models (LLMs) with symbolic programs, the authors effectively leverage the strengths of both methods: LLMs for creating interpretable natural language concepts and symbolic programs for incorporating these into decision rules. This hybrid approach not only enhances model interpretability without sacrificing performance but also makes the knowledge easily transferable to humans and adaptable for other LLMs.
Limitations are discussed in an explicit section.
In experiments, the proposed approach outperformed established baselines in multiple aspects, e.g., on OOD examples, expressiveness, and interpretability (transferability).

**Weaknesses:**

I found the manuscript somewhat challenging to follow due to the absence of consistent examples within the main text. Currently, major examples are relegated to the supplementary materials. To enhance clarity, I recommend incorporating one or two examples directly into the main manuscript and referencing them throughout.

Moreover, a clear problem statement would significantly improve comprehension. Specifically, outlining the input and output parameters before discussing the methodology would be beneficial.

Apart from these presentation issues, the experiments appear to demonstrate the efficacy of the proposed approach. However, as I feel I lack sufficient expertise in the baselines, I will defer to other reviewers for a critical assessment of the significance of these results.

Minor Points:

The structure of the document feels somewhat convoluted. There is an overuse of emphasized text formats (bold, italic, etc.), particularly noticeable on Page 3, which detracts from the core ideas being presented. I recommend retaining emphasis only on the most critical sections and reducing the use of bold and italic text.

Equation 3 could be better explained with a figure that includes a concrete example and details like $\alpha$ and $y_i$.

The text size in Figure 8 (f) makes it difficult to interpret. Consider increasing the text size for better readability.

Furthermore, the code is not submitted, making it challenging to assess reproducibility.

**Questions:**

I was not fully following the reasoning in line 268 -- 286.
Could you elaborate the following two arguments?
> Therefore, with a fixed LLM, the set of natural language prompts, ... provides a massive set of interpretable neural network modules for the task.

Why the prompts can be trivially seen as interpretable neural modules?

> ... the resulting model is inherently interpretable, as the prompt s is expressed in natural language.

Why the resulting model can be seen inherently interpretable?

---

> ### Author Response · Authors · 2024-11-25
> **Author Rebuttal Part 1/1**
>
> ---
>
> **[Original review]**
>
> *W1,W2 “I found the manuscript somewhat challenging to follow due to the absence of consistent examples within the main text. Currently, major examples are relegated to the supplementary materials. To enhance clarity, I recommend incorporating one or two examples directly into the main manuscript and referencing them throughout.” “Moreover, a clear problem statement would significantly improve comprehension. Specifically, outlining the input and output parameters before discussing the methodology would be beneficial.”*
>
> **[Response]**
>
> Thank you for these valuable suggestions! To improve clarity and readability, we have updated the main figure (Figure 2) to illustrate the entire pipeline using a simple visual classification task as an example. We hope this addresses your concern, and we would greatly appreciate any additional feedback on this update or further suggestions for improvement.
>
> ---
>
> **[Original review]**
>
> *Minor W4-7 “The structure of the document feels somewhat convoluted. There is an overuse of emphasized text formats (bold, italic, etc.), particularly noticeable on Page 3, which detracts from the core ideas being presented. I recommend retaining emphasis only on the most critical sections and reducing the use of bold and italic text.*
>
> *Equation 3 could be better explained with a figure that includes a concrete example and details like α and yi.*
>
> *The text size in Figure 8 (f) makes it difficult to interpret. Consider increasing the text size for better readability.*
>
> *Furthermore, the code is not submitted, making it challenging to assess reproducibility.”*
>
> **[Response]**
>
> Thank you for the detailed and valuable suggestions! We have incorporated your feedback into our revisions to improve the paper’s readability and presentation. Specifically, we have made the following changes:
>
> 1. **Reduced emphasis formatting:** We have minimized the use of bold and italic text throughout the paper, reserving it for highlighting only the most critical findings and conclusions.
> 2. **Enhanced explanation of Equation 3:** We updated the main pipeline figure to include Equation 3, a concrete example task, and a clear problem statement to better illustrate the entire learning process.
> 3. **Improved Figure 8(f):** We switched to a different visualization of the learned programs in Figure 8(f), which enabled us to use a much larger font for better readability.
> 4. **Code submission:** While we plan to release both the code and benchmarks upon acceptance, we provide a demo of the LSP training and testing process (though not yet organized) in the supplementary material.
>
> We hope these changes address your concerns and improve the overall clarity of the paper.
>
> ---
>
> **[Questions]**
>
> *Q1. “I was not fully following the reasoning in line 268 -- 286. Could you elaborate the following two arguments?*
>
> > *Therefore, with a fixed LLM, the set of natural language prompts, ... provides a massive set of interpretable neural network modules for the task.*
> >
>
> *Why the prompts can be trivially seen as interpretable neural modules?*
>
> > *... the resulting model is inherently interpretable, as the prompt s is expressed in natural language.*
> >
>
> *Why the resulting model can be seen inherently interpretable?”*
>
> **[Answer]**
>
> Thank you for the question. We are happy to elaborate:
>
> 1. **Why can prompts be seen as interpretable neural modules?** Pretrained (M)LLMs are powerful conditional probability models of p(y∣s,x)p(y \mid s, x)p(y∣s,x), where sss is a prompt, xxx is the input, and yyy is the output generated by the LLM based on that prompt. This allows us to implement a wide variety of neural network modules (or specialist models) simply by defining different prompts. For example:
>     1. Sentiment analysis model can be implemented by LLM(y|s=”Classify the following text into one of these categories: [positive, negative, neutral]”, x=“I like Pizzas”)
>     2. Shape detector model can be implemented as MLLM(x|s=”What is the shape of the object in this image?”, x=<image of a ball>)
> 2. **Why is the resulting model inherently interpretable?** Unlike traditional neural network modules in NSP, which are defined by numerical parameters, LLM modules are defined through natural language prompts. This makes the prompts inherently more interpretable because they are understandable to humans without requiring technical expertise in neural network representations.
>     1. To validate this claim, we conducted an empirical evaluation where human raters labeled test data by reviewing the learned rules/prompts (Table 2). The results demonstrated high “transferability” of these learned rules to humans, further supporting their interpretability.
>
> We hope this explanation helps clarifying these arguments. If you have additional questions or suggestions, we would be delighted to discuss them further!
>
> ---

---

> > ### Comment · Reviewer_ZPkh · 2024-12-02
> >
> > I thank the authors for their clarifications and apologize for my late response. I have read the rebuttal and will maintain my rating.

---

### Official Review · Reviewer_fKng · 2024-11-03

**Soundness:** 3
**Presentation:** 3
**Contribution:** 3
**Rating:** 6
**Confidence:** 4

**Summary:**

The authors in this paper introduce LLM-based Symbolic Programs (LSPs), combining Large Language Models with symbolic programming to create predictive models that balance expressiveness and interpretability. LSPs use LLMs to generate interpretable modules from raw data. With a step-by-step program generation algorithm, the proposed method could generate the symbolic programs. Besides, the authors propose a new benchmark, called IL-Bench, for evaluating interpretable learning tasks. In their evaluation, LSPs outperform traditional methods, offering interpretable, transferable, and generalizable results.

**Strengths:**

1. A new benchmark for interpretable learning tasks, for both vision and text scenarios.
2. Design a LLM-symbolic program generation algorithm (called LSP by authors) by a program structure search procedure.
3. Evaluation shows that the LSP significantly outperforms the baselines in the authors' evaluation settings.

**Weaknesses:**

1. As the authors say, LLM's responsibility is to make decisions in each LLM module in their llm-symbolic programs. Thus, generating a correct LSP relies on the LLM's ability. I am curious whether the proposed method could correspond with other open-source LLMs that could be deployed locally.

**Questions:**

Referring to W1, the author may run an additional evaluation of the proposed method with specific open-source LLMs (e.g. LlaMA3.1-8B, LlaMA3.2) that can be run locally. No matter the results, the authors could discuss some potential limitations to making LSP work with locally deployed LLMs.

The authors claim that the existing LLM benchmark could measure (M)LLM’s IL ability: "all these tasks are all zero-shot solvable, allowing LLMs to make predictions without additional rule learning. Therefore, these benchmarks are unsuitable for evaluating IL tasks."
My other questions are:

1. Are datasets from AIPS or/and AlphaGeometry, which contains IMO level mathematical problems, able to be IL benchmark?
The authors may discuss the difference between the proposed dataset and the datasets like AIPS or AlphaGeometry in terms of measuring interpretable learning ability.

FYI, AIPS aims to address inequality math problems, and AlphaGeometry aims to address geometrical math problems. Both papers propose a dataset and a problem-solving algorithm. From my point of view, the algorithm is a neural-symbolic method and is similar to LSP, which uses LLM as a module to generate a one-step decision, add a construct for geometry problems, and transform an inequality. Then, designing a high-level search problem to solve the complex problem.

2. Instead of constructing LSP, does SFT or some RL could be applied to LLM to improve the LLM's IL ability?

---

> ### Author Response · Authors · 2024-11-25
> **Author Rebuttal Part 1/1**
>
> ---
>
> **[Original review]**
>
> *W1 “As the authors say, LLM's responsibility is to make decisions in each LLM module in their llm-symbolic programs. Thus, generating a correct LSP relies on the LLM's ability. I am curious whether the proposed method could correspond with other open-source LLMs that could be deployed locally.”*
>
> **[Response]**
>
> Thank you for raising this point. With the recent advancements in open-source LLMs, we have observed that LSP performs well even when using smaller, open-source models as the learner LLM. To explore this, we conducted experiments using the same tasks as our ablation study. The results, summarized in the table below, show that LSP maintains strong performance compared to other prompt optimization methods, even with open-source LLMs.
>
> This robustness is largely attributed to the divide-and-conquer strategy of LSP:
>
> - If an LLM generates an incorrect rule at a node, the error can often be corrected by the child nodes.
> - Additionally, as the tree structure is traversed, the data distribution is divided among the nodes. This means each node is responsible for predicting only a subset of the data, simplifying the task of generating correct rules.
>
> We believe these characteristics make LSP particularly well-suited for deployment with open-source LLMs.
>
> |  | DT-Hard | Waxwing | Waterthrush |
> | --- | --- | --- | --- |
> | LSP (GPT3.5) | 96.83 (0.85) | 65.83 (4.17) | 62.50 (0.83) |
> | LSP (Gemma-2-9b-it) | **92.67 (1.65)** | **60.00 (7.58)** | **65.56 (3.14)** |
> | APE (Gemma-2-9b-it) | 59.00 (4.30) | 45.00 (0.00) | 45.00 (2.36) |
> | APO (Gemma-2-9b-it) | 90.33 (4.37) | 58.89 (8.31) | 55.56 (4.78) |
> | PromptAgent (Gemma-2-9b-it) | 85.83 (9.44) | 55.00 (1.36) | 61.67 (1.36) |
>
> (Due to license policy, we are not able to use llama in our work, instead, we adopt Gemma-2-9b-it as the learner LLM, the inference LLM is still GPT3.5.)
>
> ---
>
> **[Remaining Questions]**
>
> *Q2. “Are datasets from AIPS or/and AlphaGeometry, which contains IMO level mathematical problems, able to be IL benchmark? The authors may discuss the difference between the proposed dataset and the datasets like AIPS or AlphaGeometry in terms of measuring interpretable learning ability.”*
>
> **[Answer]**
>
> Thank you for introducing these datasets! Similar to other math datasets like GSM8K, both AIPS and AlphaGeometry consist of tasks that are zero-shot solvable. This is primarily because:
>
> - Their inputs (i.e., the problem statements) already provide all the information necessary for an LLM to solve the tasks.
> - As a result, LLMs can solve these problems without needing to learn any rules from labeled data.
>
> While the methods used in these datasets are inspiring, they differ from LSP in that they do not focus on learning interpretable predictive rules from labeled data. As the reviewer may know, Symbolic AI encompasses a broad range of problems and methodologies. For example:
>
> - Some approaches, such as those referenced in the review, use search-based structural decoding for mathematical reasoning.
> - Others equip LLMs with symbolic solvers (109).
>
> However, these methods are designed for zero-shot solvable tasks and are not directly compatible with interpretable learning (IL). We’ve added a reference to these works in Section 2 (line 99), where we discuss why certain benchmarks and approaches are not suitable for IL due to zero-shot solvability.
>
> *Q3. “Instead of constructing LSP, does SFT or some RL could be applied to LLM to improve the LLM's IL ability?”*
>
> **[Answer]**
>
> This is a great point! We agree that there are opportunities to enhance the IL ability of LLMs through methods like SFT or RLHF. Our observations suggest:
>
> - Similar to other abilities, stronger LLMs generally perform better on IL tasks.
> - As you mentioned, incorporating IL datasets during SFT could further improve IL capabilities, much like how LLMs are fine-tuned on math datasets to enhance their reasoning skills.
>
> We believe this is a promising direction for future exploration.
>
> ---

---

> > ### Comment · Reviewer_fKng · 2024-12-03
> >
> > Thank you for your detailed response. Sorry for the late reply. Based on the current implementation of the proposed LSP on the datasets, I will keep my rating. Thank you.

---

### Official Review · Reviewer_44Us · 2024-11-04

**Soundness:** 3
**Presentation:** 3
**Contribution:** 2
**Rating:** 5
**Confidence:** 4

**Summary:**

The paper proposes a new framwork, named LSP, for LLM-Symbolic Program combining LLMs with Neurosymbolic programs (NSPs). The main idea of the paper is to consider a prompted LLM as an interpretable unit, enabling the use of NSPs with a minimal domain specific language and thus narrowing this important bottleneck of NSPs. The approach is demonstrated on a proposed Interpretable-Learning benchmark. It includes a comparison with SOTA NSPs as well as some ablation study on the components of the approach. It also include an human evaluation protocol to access the interpretability of the approach.

**Strengths:**

The main strengths of the paper are :
+ **Clarity** : the paper is well written with a clear and well motivated idea. The paper provides nice illustration that enable a clear understanding of the proposed approach. The addressed research question are also very clear.  Moreover, a thorough description of the proposed framework, including ablation studies and code for reproducibility is provided.
+ **Originality** : the main originality of the paper is to use prompting and the ability of summarization of LLMs, a method named RuleSum, to obtain concrete rules and to prevent from manually designing operators, one of the bottleneck of NSPs. It is a simple approach but opening the way for LLM-based symbolic programs. The IL-bench is also an interesting contribution of the paper.
+ **Quality** : The paper is of sufficient quality for the ICLR conference standard.
+ **Significance** : the paper propose a new approach for interpretable learning, i.e. interpretable and comprehensible decision making process which an important area for XAI and trust in AI.

**Weaknesses:**

While the core idea is interesting, the paper has several limitations:
+ a first small concerns is about the title of the paper which is for me not really aligned on the content. The authors use LLMs as learnable building blocks for NSPs but are not interpretable as their own. The title of the paper should be slightly changed to better reflect the true claims of the paper.
+ **lack of clear definition of the notion of interpretability and of the underlying assumptions** : as the authors tackle the trade-off between expressiveness and interpretability, a clear definition of them should be given. For instance for me, using a minimalist DSL means less expressivity since there are only two operators. The assumption that relying on natural language brings interpretability should also be discussed. The interpretability seems to be linked to the interpretability of the prompts.
+ a better positioning to recent approach on XAI, in particular concept-based XAI, mentioned in appendix A.1 and Mechanistic Interpretability.
+ **Generality of the findings**  : the IL-Bench should be more motivated. It is not clear for me what a symbolic task is exactly. Moreover, a large part of the task are classification tasks. What about other tasks regarding the proposed approach ?
+ What is $Dist$ in equation 1 ?

**Questions:**

Some questions are in the Weaknesses section.
Other questions :
+ Could the following claim could be discussed : the resulting model is inherently interpretable as the prompt s is expressed in natural language ?
+ How this idea of using LLMs to transforms signals into high-level concepts, how the approach compared to concept induction approach ? To  Concept-based Explanations for LLMs ?
+ In the following approach, is the decision making process limited to conditional branching and tree-based models ? Is it a strong limitation ?
+ Can hallucinations can occur for RuleSum ?

---

> ### Author Response · Authors · 2024-11-25
> **Author's Response Part 1/2**
>
> ---
> **[Original review]**
>
> *W1,W2,Q1 “a first small concerns is about the title of the paper which is for me not really aligned on the content. The authors use LLMs as learnable building blocks for NSPs but are not interpretable as their own. The title of the paper should be slightly changed to better reflect the true claims of the paper. lack of clear definition of the notion of interpretability and of the underlying assumptions : as the authors tackle the trade-off between expressiveness and interpretability, a clear definition of them should be given. For instance for me, using a minimalist DSL means less expressivity since there are only two operators. The assumption that relying on natural language brings interpretability should also be discussed. The interpretability seems to be linked to the interpretability of the prompts.”*
>
> **[Response]**
>
> We appreciate the reviewer’s insights, and also agree that interpretability can take on multiple forms in the XAI literature. In this paper, we adopt a strong form of interpretability: whether a human with no prior domain knowledge can understand and apply the decision rules learned by the model (line 035).
>
> - This definition emphasizes the transfer of knowledge from AI to humans.
> - In the LSP framework, one LLM acts as the learner (optimizer), summarizing rules from data. These rules, presented in natural language, are inherently interpretable to humans. For this reason, we view LLMs as interpretable learners.
> - We empirically evaluated the interpretability of LSP by asking human raters to label test data based on the learned rules. The results showed high “transferability” of the learned rules to humans (Table 2, Last Group).
>
> We add this as a formal definition block in the revision. Please feel free to let us know your thoughts on the updated framing or if there are further suggestions for improving it!
>
> Moreover, since LLM module in the DSL can implement a wide range of operators with different prompts, it makes the minimalist DSL much more expressive than NSP. In fact, this is an advantage of LSP:
>
> - In NSP, the functions in DSL needs to be pre-defined by humans, whereas the LLM modules in LSP are fully learned from data, making the DSL more flexible and expressive.
> - Having a minimalistic DSL also makes the program structure search much more efficient, as each node has fewer functions to choose from.
>
> ---
>
> **[Original review]**
>
> *W3,Q2 “*a better positioning to recent approach on XAI, in particular concept-based XAI, mentioned in appendix A.1 and Mechanistic Interpretability.*”* “How this idea of using LLMs to transforms signals into high-level concepts, how the approach compared to concept induction approach ? To Concept-based Explanations for LLMs “
>
> **[Response]**
>
> Thank you for the suggestion! As you noted, the related work provides a literature review of concept-based methods; Following your suggestion, we further clarify how LSP compares to concept-based XAI:
>
> - **Connection**: Both concept-based methods and LSP aim to provide insights into the decision-making process.
> - **Differences**: Intrinsic concept-based methods add a “concept layer” (typically a linear transformation) to the neural network, requiring the model to make predictions based on predefined concepts. Post-hoc methods align human-defined concepts with a pretrained model. Both approaches share the same limitations: (1) they rely on human-defined high-level concepts, and (2) concepts do not fully represent the decision rules. In contrast, LSP directly learns all interpretable modules (LLM prompts) from data without relying on human prior knowledge. Furthermore, LSP fully reveals its decision process through learned prompts and program structure, while concept-based methods only partially expose the decision process.
>
> We’ve incorporated the above discussion in the Appendix A.1.
>
> ---

---

> > ### Comment · Reviewer_44Us · 2024-11-27
> > **Replying concerning the interpretability of rules**
> >
> > Thank you for your response and apologies for the late reply.
> > I still suggest a deeper discussion on that point "These rules, presented in natural language, are inherently interpretable to humans". The interpretability of a set of rules depends on a set of factors: their numbers, their lengths, and their redundancy. A set of works have been published in the XAI literature. See for instance the works on Decision Rule Sets

---

> > > ### Author Response · Authors · 2024-12-02
> > > **Further reply: Delve deeper into interpretability of LSP**
> > >
> > > Thank you for the reply and apologize for the late reply!
> > >
> > > *Q1. “Replying concerning the interpretability of rules”*
> > >
> > > **[Answer]**
> > >
> > > Thank you for the suggestion to evaluate the interpretability of rule-based models on factors such as the number of rules, their lengths, and redundancy. We agree that these structural aspects provide invaluable insights for analyzing the learned programs.
> > >
> > > To address this, we select the DT-Hard dataset for analysis, as it involves the most complex oracle generation rules. Building on the reviewer’s suggestion, we incorporated the following metrics to evaluate the complexity of the learned programs (Lakkaraju et al. 2016, Ribeiro et al. 2018):
> > >
> > > 1. **Number of rules**: Defined as the count of complete "if clauses," with "else if" and "else" counted as separate rules.
> > > 2. **Average rule length**: Measured by the average number of conditions in each rule.
> > > 3. **Redundancy**: Adapted for structured rule sets (e.g., trees in LSP). A rule is considered redundant if removing it does not significantly affect predictions. Specifically, we delete each rule (not node, one node can contain multiple rules) in the learned LSP and measure the change in test accuracy. A rule is deemed redundant if the absolute accuracy change is below a threshold. We set the threshold to 1% to account for randomness in GPT responses.
> > >
> > > The results are summarized in the following table. We highlight the following key findings:
> > >
> > > 1. Across all models, the number of rules generated is comparable, but larger models tend to produce slightly more rules than smaller ones.
> > > 2. Larger and more capable GPT models create rules with more conditions, leading to longer average rule lengths compared to smaller models like Gemma-2-9b.
> > > 3. Surprisingly, no redundant rules were identified in the LSPs learned by the models. Removing any rule resulted in a significant drop in test accuracy, indicating that all rules are required to recover the performance.
> > >
> > > To ensure reproducibility, we will include the code for computing these metrics in our final code release. We will also incorporate these discussions in the manuscript. If anything comes up, please feel free to let us know and we are happy to discuss further!
> > >
> > > | Learner LLM | test accuracy | number of rules | avg rule length | redundancy ratio |
> > > | --- | --- | --- | --- | --- |
> > > | gpt-3.5-turbo | 96.5 | 10 | 2.3 | 0% |
> > > | gpt-4 | 98.5 | 12 | 2.1 | 0% |
> > > | gemma-2-9b-it | 94.5 | 11 | 1.73 | 0% |

---

> ### Author Response · Authors · 2024-11-25
> **Author's Response Part 2/2**
>
> ---
> **[Original review]**
>
> *W4 “***Generality of the findings** : the IL-Bench should be more motivated. It is not clear for me what a symbolic task is exactly. Moreover, a large part of the task are classification tasks. What about other tasks regarding the proposed approach ?*”*
>
> **[Response]**
>
> Thank you for highlighting this point. We provide an overview of all tasks in Appendix Table 8. The symbolic tasks are decision-making problems where all variables, values, and labels are represented using abstract symbols (e.g., x1, A1, foo, bar). This design has several advantages:
>
> 1. **Ensures non-zero-shot solvability:** Since the tasks use abstract symbols, LLMs cannot rely on prior knowledge to make predictions. For instance, if the task had real-world context, such as predicting rainfall based on humidity, the LLM can make predictions even without learning any underlying rules from data.
> 2. **Scalability:** Symbolic tasks can be automatically generated and scaled to any size or complexity.
> 3. **Known oracle rules:** The underlying rules used for data generation are explicitly known, which facilitates model diagnostics. For example, in Appendix A.3, we are able to compare the learned LSP to the oracle rules.
>
> Consistent with traditional XAI approaches, particularly those focused on learning interpretable decision-making models (e.g., NSP), our work primarily focuses on predictive tasks. However, the versatility of the LSP framework, powered by LLM modules, allows it to be extended to any task solvable by LLMs. We consider this an exciting direction for future exploration.
>
> ---
>
> **[Original review]**
>
> *W5 “*What is Dist in equation 1 ?*”*
>
> **[Response]**
>
> Thank you for bringing it up. Dist(a1, a2) is the distance function between a1 and a2.
>
> ---
>
> **[Remaining Questions]**
>
> *Q3 “In the following approach, is the decision making process limited to conditional branching and tree-based models ? Is it a strong limitation ?”*
>
> **[Answer]**
>
> Thank you for raising this point! The resulting program adopts a tree structure, not due to a constraint of conditional branching, but because trees are a widely used representation for programs (e.g., any code can be represented into a syntax tree). This alignment also explains why prior NSP methods naturally discover tree-structured programs.
>
> *Q4 “Can hallucinations can occur for RuleSum ?”*
>
> **[Answer]**
>
> Yes! we have observed that the learner LLM does not always summarize the rules correctly. However, we also discover the following:
>
> - Stronger models (e.g., GPT-4) tend to exhibit significantly fewer hallucinations compared to weaker models (e.g., GPT-3.5 or gemini-medium).
> - The LSP framework, with its divide-and-conquer approach, is more tolerant to errors compared to vanilla prompt optimization algorithms. This is because errors in parent rules can often be corrected by child branches.
>
> We believe that fine-tuning LLMs to improve rule summarization is a promising direction for future research.
>
> ---

---

> > ### Comment · Reviewer_44Us · 2024-11-27
> > **Concerning W5 “What is Dist in equation 1 ?”**
> >
> > The question was more focused on the specific distance measure used in this case.

---

> > > ### Author Response · Authors · 2024-12-02
> > > **Further Reply: Dist in equation 1**
> > >
> > > Sorry for misunderstanding your original question! Equation 1 is intended as an example of a DSL to illustrate what a DSL in NSP could look like. The primary goal is to convey that DSLs in traditional NSP approaches must be manually defined. For this reason, we did not specify the exact measure for the `Dist()` function. We'd imagine that common options, such as cosine similarity or L2 distance, could be used here for `Dist()`, depending on the specific requirement of the task.

---

### Author Response · Authors · 2024-11-25
**Common Reply to all reviewers**

We sincerely thank our reviewers for the invaluable and constructive feedback, as well as for recognizing the originality of our method and benchmark (R1, R2, R3) and the performance improvements demonstrated in the empirical results (R2, R3). Below, we summarize the major updates and additional results incorporated during the rebuttal process (marked in blue in the revised paper):

1. **[Writing]** Added an explicit definition block for interpretability in Section 1.
2. **[Writing]** Reduced the overuse of text highlights (bold, italic) for better readability.
3. **[Writing]** Incorporated a comparison of LSP with concept-based XAI methods in Appendix A.1.
4. **[Writing]** Updated the main figure (Figure 2) to include a concrete example, problem statement, and DSL.
5. **[Writing]** Improved the visualization of learned programs in Figure 6 and Figure 8(f).
6. **[EXP]** Conducted an ablation study using open-source models.
7. **[Demo]** Update supplementary material with a demo code.

We hope our responses and updates address the reviewers’ questions and concerns. If there are any additional points for discussion, we would be delighted to engage further.

---

> ### Author Response · Authors · 2024-12-02
> **Let us know if you have further questions!**
>
> Dear Reviewers,
>
> Thank you again for reviewing our paper and leaving insightful suggestions! As the deadline approaches, we want to reach out to see if our rebuttal addresses your concerns? If you have any further questions, please feel free to let us know. We are happy to address any questions you might have!
>
> Best,
> LSP Authors

---

### Meta-Review · Area_Chair_YVqp · 2024-12-21

**Metareview:**

The paper presents a novel approach combining LLMs with symbolic programs to enhance interpretability in AI. It introduces LSPs, leveraging LLMs' interpretable modules and symbolic rules for decision-making.

Strengths include the innovative integration of LLMs and symbolic learning, the development of a divide-and-conquer training approach, and the introduction of IL-Bench for evaluation.

Weaknesses could be the need for more extensive testing across diverse datasets, and presentation improvement.

The decision to accept is based on the paper's innovative approach to improving AI interpretability, strong empirical results, and the potential impact on the field.

**Additional Comments On Reviewer Discussion:**

Reviewer 44Us and Reviewer ZPkh  raised the problem about the interpretability of the proposed method, which is basically solved by the authors.

---

### Decision · Program_Chairs · 2025-01-22

Accept (Poster)